# Structural basis of subtype-selective competitive antagonism for GluN2C/2D-containing NMDA receptors

Jue Xiang Wang[1,2], Mark W. Irvine [3], Erica S. Burnell[3,4], Kiran Sapkota[5], Robert J. Thatcher[3], Minjun Li[1], Noriko Simorowski[1], Arturas Volianskis [6], Graham L. Collingridge[3,7,8], Daniel T. Monaghan[5], David E. Jane[3]* & Hiro Furukawa [1,2]*

*N*-Methyl-D-aspartate receptors (NMDARs) play critical roles in the central nervous system. Their heterotetrameric composition generates subtypes with distinct functional properties and spatio-temporal distribution in the brain, raising the possibility for subtype-specific targeting by pharmacological means for treatment of neurological diseases. While specific compounds for GluN2A and GluN2B-containing NMDARs are well established, those that target GluN2C and GluN2D are currently underdeveloped with low potency and uncharacterized binding modes. Here, using electrophysiology and X-ray crystallography, we show that UBP791 ((2S*,3R*)-1-(7-(2-carboxyethyl)phenanthrene-2-carbonyl)piperazine-2,3-dicarboxylic acid) inhibits GluN2C/2D with 40-fold selectivity over GluN2A-containing receptors, and that a methionine and a lysine residue in the ligand binding pocket (GluN2D-Met763/Lys766, GluN2C-Met736/Lys739) are the critical molecular elements for the subtype-specific binding. These findings led to development of UBP1700 ((2S*,3R*)-1-(7-(2-carboxyvinyl)phenanthrene-2-carbonyl)piperazine-2,3-dicarboxylic acid) which shows over 50-fold GluN2C/2D-selectivity over GluN2A with potencies in the low nanomolar range. Our study shows that the L-glutamate binding site can be targeted for GluN2C/2D-specific inhibition.

---

[1] WM Keck Structural Biology Laboratory, Cold Spring Harbor Laboratory, Cold Spring Harbor, NY 11724, USA. [2] Watson School of Biological Sciences, Cold Spring Harbor Laboratory, Cold Spring Harbor, NY 11724, USA. [3] Glutamate Research Group, School of Physiology, Pharmacology and Neuroscience, University of Bristol, Biomedical Sciences Building, University Walk, Bristol, BS8 1TD, UK. [4] School of Chemistry, National University of Ireland Galway, Galway H91TK33, Ireland. [5] Department of Pharmacology and Experimental Neuroscience, University of Nebraska Medical Center, Omaha, NE 68198-5800, USA. [6] Centre for Neuroscience and Trauma, Blizard Institute, Barts and The London School of Medicine and Dentistry, Queen Mary University of London, London, UK. [7] Tanz Centre for Research in Neurodegenerative Diseases, Department of Physiology, University of Toronto, Krembil Discovery Tower, 60 Leonard Avenue, Toronto, ON M5T 0S8, Canada. [8] Lunenfeld-Tanenbaum Research Institute, Mount Sinai Hospital, 600 University Avenue, Toronto, ON M5G 1X5, Canada. *email: David.Jane@bristol.ac.uk; furukawa@cshl.edu

**N**-Methyl-D-aspartate receptors (NMDARs) belong to the ionotropic glutamate receptor (iGluR) family and are ligand-gated ion channels that mediate the majority of excitatory synaptic transmission in the central nervous system. NMDARs play critical roles in brain development and functions such as learning and memory and have been implicated in an array of neurological diseases and disorders, including depression, stroke, seizure, schizophrenia, Alzheimer's disease and Parkinson's disease[1,2]. Their roles have been most extensively studied as part of the postsynaptic density where NMDARs co-localize with non-NMDAR iGluRs including α-amino-3-hydroxy-5-methyl-4-isoxazolepropionic acid receptors (AMPARs) along with many other postsynaptic density proteins. Recent evidence, however, demonstrates that presynaptic and extrasynaptic NMDARs play important regulatory roles in neuronal signaling and diseases[3,4]. It has been well known that overactivation of NMDARs is associated with neuronal cell death caused by stroke, traumatic brain injury, and neurodegenerative diseases[5–7], thus leading the field to pursue means to alleviate NMDAR activities. A number of NMDAR inhibitors have been in clinical trials in the past two decades, including the open channel blocker memantine, which has been approved by the FDA to treat Alzheimer's disease[8].

NMDARs function as heterotetramers composed of GluN1 subunits (with eight splice variants 1a-4a, 1b-4b) and GluN2A/2B/2C/2D and/or GluN3A/3B subunits. GluN1 and GluN3 subunits bind glycine (Gly), whereas GluN2 subunits bind L-glutamate. Hence, GluN1/GluN2 NMDAR activation requires binding of both Gly and L-glutamate, whereas GluN1/GluN3 NMDAR activation requires only Gly. Each receptor subunit has a modular build with an amino-terminal domain (ATD), a ligand-binding domain (LBD), a transmembrane domain (TMD), and a carboxyl-terminal domain (CTD) (Fig. 1a, b). The crystal structures of the intact NMDAR structures[9,10] showed that the subunits assemble as a dimer of GluN1–GluN2 heterodimers with a swap of heterodimer partners from the ATD to the LBD layer (Fig. 1a). Importantly, previously obtained structures of isolated ATDs and LBDs[11–13] are identical to the ones observed in the intact NMDARs[9,10] demonstrating the physiological relevance of conducting structural biology on the isolated extracellular domains. All ATDs and LBDs have distinct bilobed clamshell-like architectures. In the case of LBDs, agonists (e.g. Gly and L-glutamate) or competitive antagonists (e.g. 5,7-dichlorokynurenic acid and 2-amino-5-phosphonopentanoic acid) bind to the cleft between the two lobes.

Subunit diversity is a hallmark of the NMDAR family and can be potentially exploited to target specific diseases. Different combinations of GluN1–3 subunits give rise to specific di- and tri-heteromeric NMDAR subtypes with a wide spectrum of electrophysiological and pharmacological properties. Extensive research has shown distinct spatio-temporal distribution of NMDAR subtypes in the brain[14–17] implying unique roles of different NMDAR subtypes in specific aspects of brain development and functions, and suggesting therapeutic potential for subtype-specific targeting of NMDARs. Thus, development of highly subtype-specific reagents will advance our understanding of the biological roles of NMDAR subtypes in brain functions and development, and may provide possible treatments for the aforementioned diseases and disorders. While GluN2A/2B-containing NMDARs are dominant subtypes that are expressed in the adult brain, the expression of GluN2C/2D-containing NMDARs is restricted to discrete regions critical for diseases. For example, in schizophrenia, recent evidence has pointed to critical involvement of NMDAR hypofunction in cortical GABAergic neurons[18,19] where GluN2D subunits are highly expressed[20,21]. In Parkinson's disease where over-firing of subthalamic nucleus (STN) neurons occurs due to the loss of dopaminergic neurons in the substantia nigra pars compacta, GluN2D-containing NMDARs may be the relevant target since they are present and mediate synaptic neurotransmission in the STN[22].

One of the key limitations in studying GluN2C- and GluN2D-containing NMDARs in both pre- and postsynaptic processes has been the lack of highly potent and subtype-specific agonists and antagonists. This is in stark contrast to the GluN2B- and GluN2A-containing NMDARs where highly subtype-specific compounds, ifenprodil[23] and TCN-201[24–26], respectively, are available. Still, compounds with twofold to tenfold GluN2C and GluN2D-selectivity over GluN2A/2B, such as PPDA ((2S*,3R*)-1-(phenanthrene-2-carbonyl)piperazine-2,3-dicarboxylic acid) and its derivatives UBP141 ((2R*,3S*)-1-(phenanthrene-3-carbonyl) piperazine-2,3-dicarboxylic acid) and UBP145 ((2R*,3S*)-1-(9-bromophenanthrene-3-carbonyl)piperazine-2,3-dicarboxylic acid)[27–29] were frequently used to show that presynaptic GluN2D-containing receptors contribute to the major component of short-term potentiation[30] and spike timing-dependent long-term depression[31], and mediate synaptic currents in the juvenile hippocampus[20]. While there has been much improvement in more GluN2C and/or GluN2D-specific compounds in recent years[32–36], all of these allosteric compounds have shown $IC_{50}$ values in the high nanomolar to micromolar range and still lack well-defined binding sites on the NMDAR, limiting the ability for rational compound optimization. In contrast, the first reported GluN2C/2D-specific compound with well-defined binding mode at the GluN2 LBD cleft was the competitive antagonist PPDA, with $K_i$ values in the sub-micromolar range but with only twofold to sixfold GluN2C/2D-selectivity over GluN2A/2B[27]. Screening efforts led to the discovery of those similar compounds, UBP141 and UBP145, with up to approximately tenfold GluN2C/2D selectivity[28].

In this study, we present a PPDA-derivative UBP791 (2S*,3R*)-(1-(7-(2-carboxyethyl)phenanthrene-2-carbonyl)piperazine-2,3-dicarboxylic acid) which showed 47-fold and 16-fold preference of GluN2C/2D- over GluN2A- and GluN2B-containing NMDARs, respectively, and use UBP791 to study the key molecular determinants within GluN2D that confer GluN2C/GluN2D-selective compound binding. Through X-ray crystallography and electrophysiology, we determined that a combination of a methionine and a lysine unique to GluN2C/2D (rat GluN2D-Met763/Lys766, GluN2C-Met736/Lys739) confers subtype-selective binding of UBP791. Rationally modifying UBP791 then led to a greatly improved compound, UBP1700 ((2S*,3R*)-1-(7-(2-carboxyvinyl)phenanthrene-2-carbonyl)piperazine-2,3-dicarboxylic acid), which had 63-fold and 52-fold selectivity for GluN2D (50- and 40-fold for GluN2C) over GluN2A and GluN2B, and still showed high potency with $K_i$ values in the low nanomolar range. Hence, our study demonstrates that despite high conservation of the GluN2 LBDs (with sequence identities of 69-82% in rat LBDs) particularly at the L-glutamate-binding pocket, understanding the exact binding mode of a compound like UBP791 allows exploration of the LBD as a potential subtype-specific target.

## Results

**UBP791 shows improved GluN2C/2D-selectivity over GluN2A/2B**. Our compound, UBP791, is a competitive antagonist derived from PPDA that has 2.5 to 6-fold GluN2C/2D-selectivity over GluN2A and GluN2B (Fig. 1c)[27]. The chemical definition of UBP791 used in this study is a *cis*-racemic mixture of (2S,3R)- and (2R,3S)-isomers. To verify GluN2C/2D-selectivity over GluN2A/2B of UBP791, we first determined potencies of ion channel inhibition using two-electrode voltage clamp (TEVC) electrophysiology on *Xenopus laevis* oocytes injected with cRNA encoding various GluN2 subunits in combination with GluN1-4a

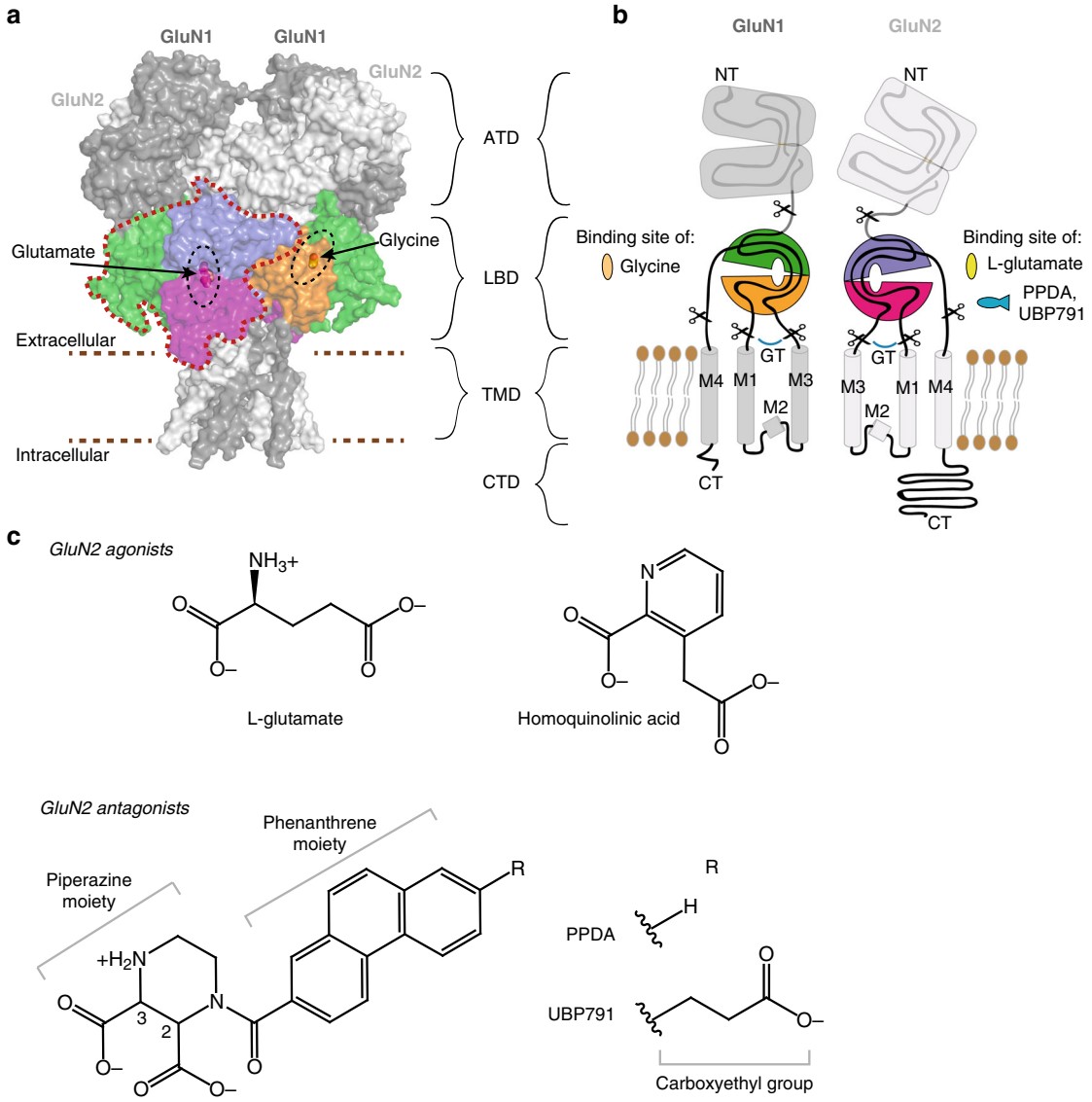

**Fig. 1 NMDAR domain organization and ligands. a** GluN1/GluN2 form tetrameric receptors with two GluN1 (dark gray) and two GluN2 (light gray) subunits. Each subunit has an ATD, LBD, TMD, and CTD domain (CTD not included). The LBD layer contains a dimer of GluN1–GluN2 heterodimers, one of which is highlighted by a surrounding red dashed line. The GluN1 upper lobe (D1) and lower lobe (D2) are colored in green and orange, and the GluN2 D1 and D2 in purple and magenta. The surface presentation is produced from the PDB structure 4PE5. **b** Schematic representation of GluN1 and GluN2 subunits. Glycine, L-glutamate, and competitive antagonists bind in the LBD clefts. The modular build of the subunits allows isolation of the GluN1 and GluN2A LBDs by replacing the M1–M3 transmembrane sequence with a Gly-Thr dipeptide linker. Color coding as in (**a**). **c** Chemical structures of GluN2 agonists L-glutamate and homoquinolinic acid and antagonists PPDA and UBP791 at pH ~7. PPDA and UBP791 are *cis*-racemic mixtures of (2*S*,3*R*)- and (2*R*,3*S*)-isomers.

(hence GluN1/GluN2 NMDARs). Towards this end, we estimated $IC_{50}$ values by measuring macroscopic currents at fixed Gly and L-glutamate concentrations and varying UBP791 concentrations (Fig. 2a, b). We also estimated $EC_{50}$ values of L-glutamate by measuring macroscopic currents at a fixed Gly concentration (100 μM) and varying L-glutamate concentrations (Fig. 2c). Inhibition potencies are defined as $K_i$ values, which are derived from the Cheng–Prusoff equation[37] that considers $EC_{50}$, $IC_{50}$, and the L-glutamate concentration used in the experiments. The $K_i$ values for the GluN1/GluN2C and GluN1/GluN2D NMDARs were 80–90 nM, which were about 16- to 17-fold and 47- to 50-fold lower than those for the GluN1/GluN2B and the GluN1/GluN2A NMDARs, respectively (Fig. 2d). Thus, UBP791 represents a substantial improvement from PPDA[27].

**Key elements for subtype-specific agonist/antagonist binding.** Our previous structure of the GluN1–GluN2A LBD complexed to PPDA unambiguously mapped the antagonist-binding site at the cleft of the bilobe architecture of the GluN2 LBD (PDB: 4NF6). Given the structural similarity between PPDA and UBP791, we hypothesized that the binding mode of UBP791 is similar to that of PPDA and that the molecular determinants for GluN2C/2D-selective binding of UBP791 likely reside within the ligand-binding pocket. In this study, we focus on the comparison between the GluN2A and GluN2D subunits, since sequence and functional similarities between GluN2A and GluN2B and between GluN2C and GluN2D are high. Near the PPDA-binding pocket in the GluN2A LBD (Fig. 3a), the primary sequences across the GluN2A-D subunits are mostly identical except for

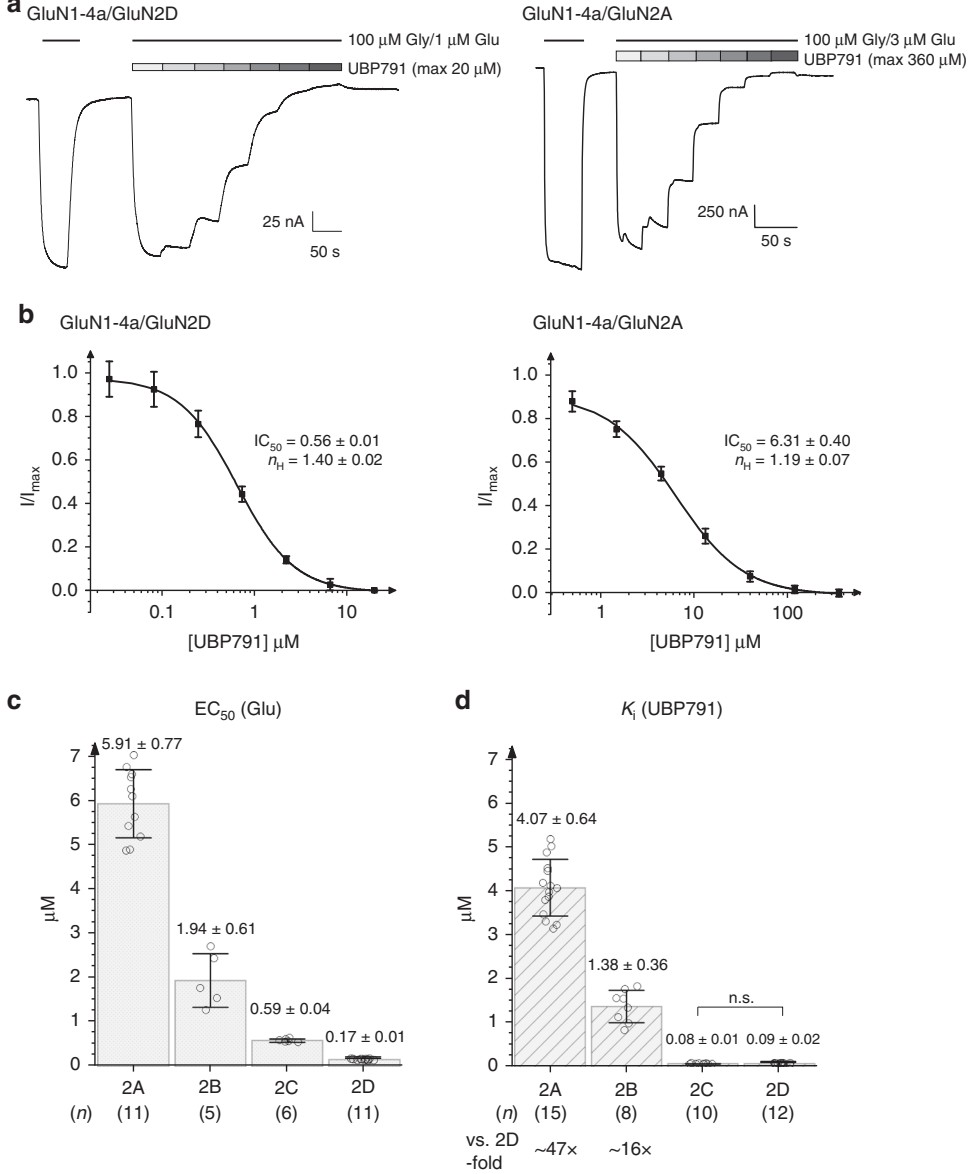

**Fig. 2 UBP791 binds preferentially to GluN2C/2D-containing NMDARs over GluN2A/2B-containing NMDARs. a** Representative TEVC dose-response traces of GluN1-4a/GluN2D (left panel) or GluN1-4a/GluN2A (right panel) NMDARs held at −60 mV. Currents were evoked by application of 100 μM glycine and 1 or 3 μM L-glutamate and inhibited by varying concentrations of UBP791 (threefold increments to max. 20 and 360 μM as shown). **b** Averaged dose-response curves (mean ± s.d.) for inhibition with UBP791 from ten GluN1-4a/GluN2D and 15 GluN1-4a/GluN2A recordings fit with the Hill equation to calculate $IC_{50}$ and Hill coefficient ($n_H$). **c** $EC_{50}$ values for GluN1-4a/GluN2 (**a–d**) obtained from L-glutamate dose-response curves. **d** Inhibition potency ($K_i$) values calculated using the Cheng–Prusoff equation with the determined $EC_{50}$ and $IC_{50}$ values. Single data points are shown as open circles, the bar graph represents the mean with error bars for s.d., the number of recordings (n) and the fold difference to the $K_i$ of GluN2D are as shown. Pairwise comparison shows subtypes have different potencies ($p < 0.05$ with two-tail t test and Bonferroni correction) except where stated (n.s.).

four residues, GluN2A-Ala414, Lys738, Gly740, and Arg741, which are Arg, Met, Arg, and Lys residues in GluN2D, respectively (Fig. 3a; yellow residues and arrows in the sequence alignment). GluN2A-Ala414 is located in the D1-lobe, whereas GluN2A-Lys738, Gly740, and Arg741 are on Helix H in the D2-lobe proximal to the predicted binding site for the carboxyethyl group of UBP791. These residues are completely conserved among human, rat, and chicken, though GluN2A-Ala414 and -Lys738 are not conserved in *X. laevis* (frog) and *Danio rerio* (zebrafish) (Supplementary Fig. 1). To assess the involvement of the above four residues in defining GluN2A and GluN2D subtype-specific pharmacological properties, we incorporated the

mutations, Ala414Arg, Lys738Met, Gly740Arg, and Arg741Lys, in the full-length GluN2A (GluN2A-4m), and measured excitatory potency of L-glutamate and inhibitory potency of UBP791 by TEVC as above. The GluN2A-4m showed drastically increased L-glutamate potency ($EC_{50} = 0.61 \pm 0.07$ μM) compared with GluN2A ($EC_{50} = 5.91 \pm 0.77$ μM) although not to the equal level of GluN2D ($EC_{50} = 0.17 \pm 0.01$ μM) (Fig. 3c, Supplementary Fig. 2a, b). It also showed a large increase in UBP791 potency ($K_i = 0.71 \pm 0.14$ μM) compared with GluN2A ($K_i = 4.07 \pm 0.64$ μM) although not to the equal level of GluN2D ($K_i = 0.09 \pm 0.02$ μM) (Fig. 3d, Supplementary Fig. 2c, d). Overall, the GluN2D-like potencies for L-glutamate and UBP791 in the

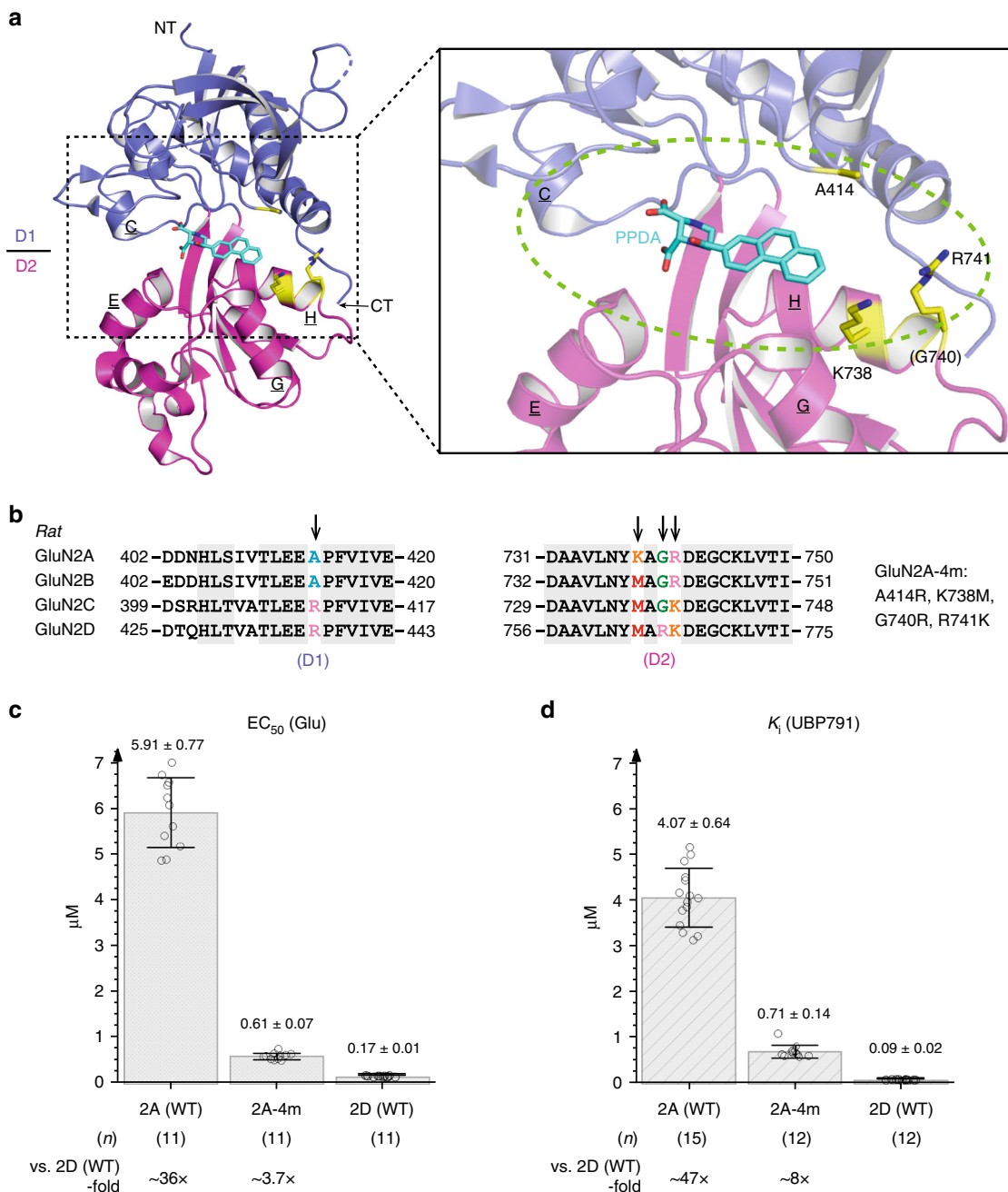

**Fig. 3 Site-directed mutagenesis of GluN2A binding pocket and its effect on L-glutamate and UBP791 sensitivity. a, b** The GluN2A LBD of the GluN1–GluN2A LBD crystal structure complexed to PPDA (cyan sticks; PDB code: 4NF6) colored as in Fig. 1. Residues within the ligand-binding pocket (green dashed oval) which are not conserved among GluN2A-D in the sequence alignment (panel b arrows) are shown in yellow. Four mutations (Ala414Arg, Lys738Met, Gly740Arg, Arg741Lys) were introduced in GluN2A to generate GluN2A-4m. **c** $EC_{50}$ values for L-glutamate and **d** $K_i$ values for UBP791 for GluN1-4a/GluN2A-4m were obtained by TEVC. Dose-response for UBP791 was measured in the presence of 100 μM glycine and 1 μM L-glutamate with varying UBP791 concentrations (Supplementary Fig. 2). Single data points are shown as open circles, the bar graph represents the mean with error bars for s.d., and the number of recordings (n) and the fold-difference to $EC_{50}$ and $K_i$ of GluN2D are as shown. Pairwise comparison shows subtypes have different potencies ($p < 0.0001$ with two-tail $t$-test and Bonferroni correction).

GluN2A-4m mutant indicated that the four residue positions in the binding cleft (Fig. 3a) are the major determinants of subtype-specific ligand binding.

**GluN2A-4m LBD serves as structural proxy for GluN2D LBD.** To understand the exact-binding mode of UBP791 and the molecular basis for the GluN2C/2D subtype-selectivity, and to determine unexplored possibilities for compound development,

we next sought to obtain structures of the GluN2A LBD and GluN2D LBD complexed to UBP791. While crystal structures of GluN2D LBD in complex with agonists and partial agonists have been successfully obtained[38,39], an antagonist-bound GluN2D LBD structure has been technically difficult to capture. The only competitive antagonist-bound crystal structures of NMDAR GluN2 LBDs to date are the ones complexed to GluN1–GluN2A LBDs[40–42], which have been obtained by soaking Gly- and L-glutamate-bound GluN1–GluN2A LBD crystals against the

**Table 1 Data collection and refinement statistics (molecular replacement).**

| | GluN1/GluN2A-4m LBD +Gly, Glu | GluN1/GluN2A LBD +Gly, UBP791 | GluN1/GluN2A-4m LBD +Gly, UBP791 |
|---|---|---|---|
| Data collection | | | |
| Space group | $P2_12_12_1$ | $P2_12_12_1$ | $P2_12_12_1$ |
| Cell dimensions | | | |
| $a, b, c$ (Å) | 54.64, 90.06, 125.15 | 59.37, 85.75, 119.87 | 58.99, 85.08, 120.41 |
| $\alpha, \beta, \gamma$ (°) | 90, 90, 90 | 90, 90, 90 | 90, 90, 90 |
| Resolution (Å) | 73.10–1.66 (1.69–1.66) | 69.74–2.52 (2.56–2.52) Staraniso: 69.74–2.13 (2.32–2.13) | 49.15–2.41 (2.49–2.41) |
| $R_{merge}$ | 0.073 (0.515) | 0.104 (0.903) Staraniso: 0.117 (1.318) | 0.156 (0.668) |
| $I/\sigma I$ | 14.3 (2.2) | 12.7 (2.3) Staraniso: 10.5 (1.5) | 11.60 (1.89) |
| Completeness (%) | 96.40 (74.9) | 99.9 (100.0) Star spherical/ ellipsoidal: 78/94.3 (19.7/61.3) | 98.03 (84.68) |
| Redundancy | 6.3 (3.9) | 6.6 (6.8) Star: 6.6 (6.6) (staraniso file) | 5.8 (4.6) |
| Refinement | | | |
| Resolution (Å) | 50.07–1.66 | 69.74–2.13 | 49.15–2.41 |
| No. reflections | 71,406 (5,481) | 26,485 (178) | 23,716 (1,995) |
| $R_{work}/R_{free}$ | 0.1794/0.2118 | 0.2075/0.2531 | 0.1914/0.2453 |
| No. atoms | | | |
| Protein | 4,553 | 4,442 | 4,547 |
| Ligands | 15 | 38 | 38 |
| Water | 607 | 66 | 115 |
| $B$-factors (Å$^2$) | | | |
| Protein | 30.51 | 55.39 | 40.89 |
| Ligands | 16.00 | 53.10 | 35.08 |
| Water | 40.08 | 53.50 | 38.24 |
| R.m.s. deviations | | | |
| Bond lengths (Å) | 0.009 | 0.002 | 0.002 |
| Bond angles (°) | 0.997 | 0.49 | 0.48 |

Datasets for all of the three structures above were collected from single crystals. Values in parentheses are for highest-resolution shell

crystallization buffer containing Gly and GluN2 antagonists to substitute L-glutamate with the antagonists within the GluN2A LBD (see "Methods"). Similarly soaking the agonist-bound GluN2D LBD crystals[38] or co-crystallization with antagonists did not result in antagonist-bound GluN2D LBD structures. We therefore attempted to use GluN2A-4m LBD as a structural mimic of GluN2D LBD and a tool to capture antagonist binding by GluN2D. Here, we found that GluN2A-4m LBD protein can be recombinantly expressed, purified, and co-crystallized with GluN1 LBD protein in the presence of Gly and L-glutamate as in the case of GluN1–GluN2A LBD heterodimers[40,41]. The crystals of the Gly- and L-glutamate-bound GluN1–GluN2A-4m LBDs showed X-ray diffraction to 1.7 Å (Table 1) and the structure was solved by molecular replacement using the Gly/glutamate-bound GluN1–GluN2A LBD heterodimer (PDB: 4NF8)[40] as search probe. The asymmetric unit contained one GluN1–GluN2A-4m LBD heterodimer assembled in the back-to-back orientation (Fig. 4a) nearly identical to that observed in the previous GluN1–GluN2A LBD structures[12,40–43]. GluN2A-4m and GluN2D LBDs had nearly identical overall fold with a minor difference in the Loop1 region (Fig. 4b). More importantly, residues around the L-glutamate binding site are in similar orientations to those of GluN2D (RMSD 0.115 Å over 7 residues/ 53 atoms) (Fig. 4c). Only minor differences in side chain orientations of GluN2A-4m Arg740 and Lys741 were observed compared with the equivalent residues in GluN2D (GluN2D-Arg765 and Lys766) (Fig. 4d). Hence, the structural comparison verified

that the GluN2A-4m LBD serves as a valid GluN2D LBD structural proxy suitable for studying binding of agonists and competitive antagonists.

**UBP791-bound GluN1–GluN2A and GluN1–GluN2A-4m LBD structures.** To identify the molecular determinants underlying GluN2C/2D-specific binding of UBP791 over GluN2A/2B, we next sought to obtain and compare structures of GluN1–GluN2A LBD and GluN1–GluN2A-4m LBD complexed to UBP791. Toward this end, we implemented the strategy of substituting antagonists into the agonist-bound GluN1–GluN2A LBD crystals by a soaking method similar to our previous studies[40,41]. The crystals of the UBP791-soaked GluN1–GluN2A LBD conferred X-ray diffraction that extended to the Bragg spacing of 2.5 Å (Table 1). The crystal structure was solved by using the PPDA-bound GluN1–GluN2A LBD (PDB: 4NF6)[40] as search probe. In the asymmetric unit, the interaction between the GluN1 and GluN2A LBDs is dominated at the D1 lobes as in the agonist-bound GluN1–GluN2A LBD (Fig. 5a). However, the D2 lobe of the UBP791-bound GluN2A LBD 'opens' by 22.1° similar to the extent previously observed in the PPDA-bound GluN1–GluN2A LBD (20.4°; Supplementary Fig. 3a, b)[40]. This large conformational change in the UBP791-bound GluN2A LBD compared with the agonist-bound form is reflected in changes in the unit cell dimensions (Table 1). At the D1–D2 bilobe cleft, clear density for the PPDA-backbone of UBP791 is present. However, the

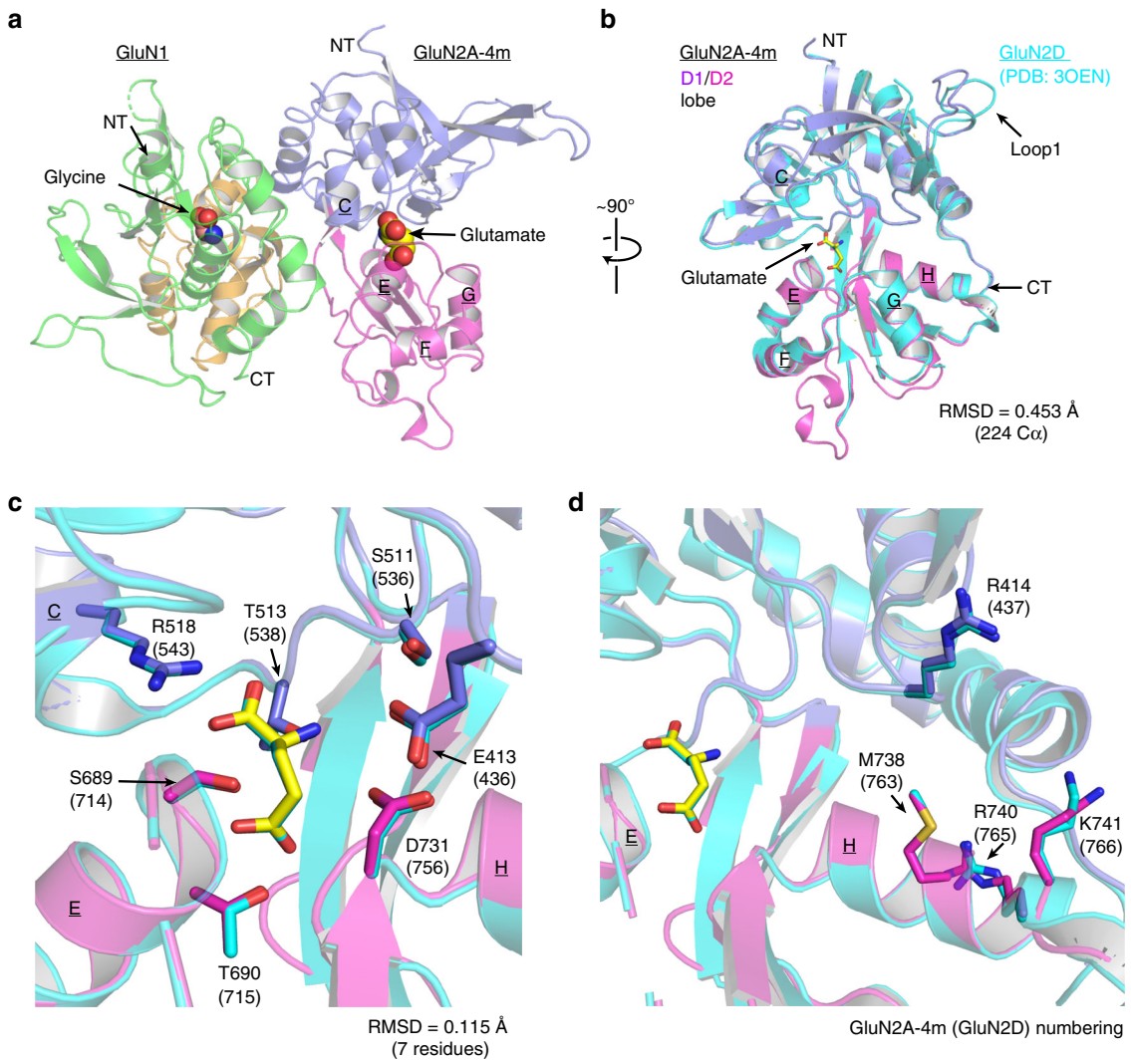

**Fig. 4 Crystal structure of GluN1–GluN2A-4m LBD and comparison with GluN2D LBD. a** Crystal structure of the GluN1–GluN2A-4m LBD heterodimer in complex with glycine (orange sphere) and L-glutamate (yellow sphere). Helices are labeled with underlined letters and the color coding is in accordance to Fig. 1. **b** Structures of the GluN2A-4m LBD and the GluN2D LBD (PDB: 3OEN) can be superposed with one another at an RMSD of 0.45 Å over 224 Cα positions. **c** Comparison of the L-glutamate-binding sites of GluN2D and GluN2A-4m shows the similar mode of ligand-receptor interactions. Residue numbering shown for GluN2A-4m and GluN2D (in brackets). **d** Mutated residues in GluN2A-4m overlap well with the equivalent residues in GluN2D with minor differences in side chain orientations.

density of the carboxyethyl group is broken and disordered (Fig. 5a).

The crystal structure of the GluN1–GluN2A-4m LBD was obtained by implementing a modified agonist co-crystallization and soaking protocol. This involved co-crystallizing GluN1–GluN2A-4m LBD with Gly and the low potency agonist homoquinolinic acid (HQA), followed by soaking against a buffer containing UBP791 in the presence of Gly. This modification resulted in the structure fully occupied by UBP791 at 2.4 Å resolution (Fig. 5d). The extent of the bilobe 'opening' of the GluN2A-4m LBD as a result of UBP791 binding was 22.3° (Supplementary Fig. 3c) similar to that of the GluN1–GluN2A LBD (Supplementary Fig. 3b). However, the electron density for the UBP791 ligand in this structure is fully continuous and substantially more ordered suggesting favorable binding of UBP791 to GluN1–GluN2A-4m LBD over GluN1–GluN2A LBD (Fig. 5d). En route to the above UBP791-bound structures, we have also obtained structures of the GluN1–GluN2A LBD and the GluN1–GluN2A-4m LBD in complex with HQA. These structures explained the underlying

mechanism for the low potency nature of HQA and are presented in Supplementary Note 1.

**Molecular elements for subtype-selective binding of UBP791.** Structural comparison between the GluN1–GluN2A LBD and the GluN1–GluN2A-4m LBD complexed to UBP791 provided important insights into preferential binding of UBP791 to GluN2D over GluN2A. In both crystal structures, only the (2S,3R)-enantiomer of UBP791 was observed in the inter-D1–D2 domain cleft, consistent with the previous finding that (2S,3R)-PPDA has approximately tenfold higher potency compared with (2R,3S)-PPDA[29,40] and about 1.4-fold higher potency than the racemate[29], suggesting that the more potent isomer is able to bind more effectively particularly in the lower concentration range compared with the less potent isomer. The binding modes of the piperazine and phenanthrene moieties are identical to those observed in the structure of the GluN1–GluN2A LBD complexed to PPDA[40]. That is, the piperazine ring interacts via polar

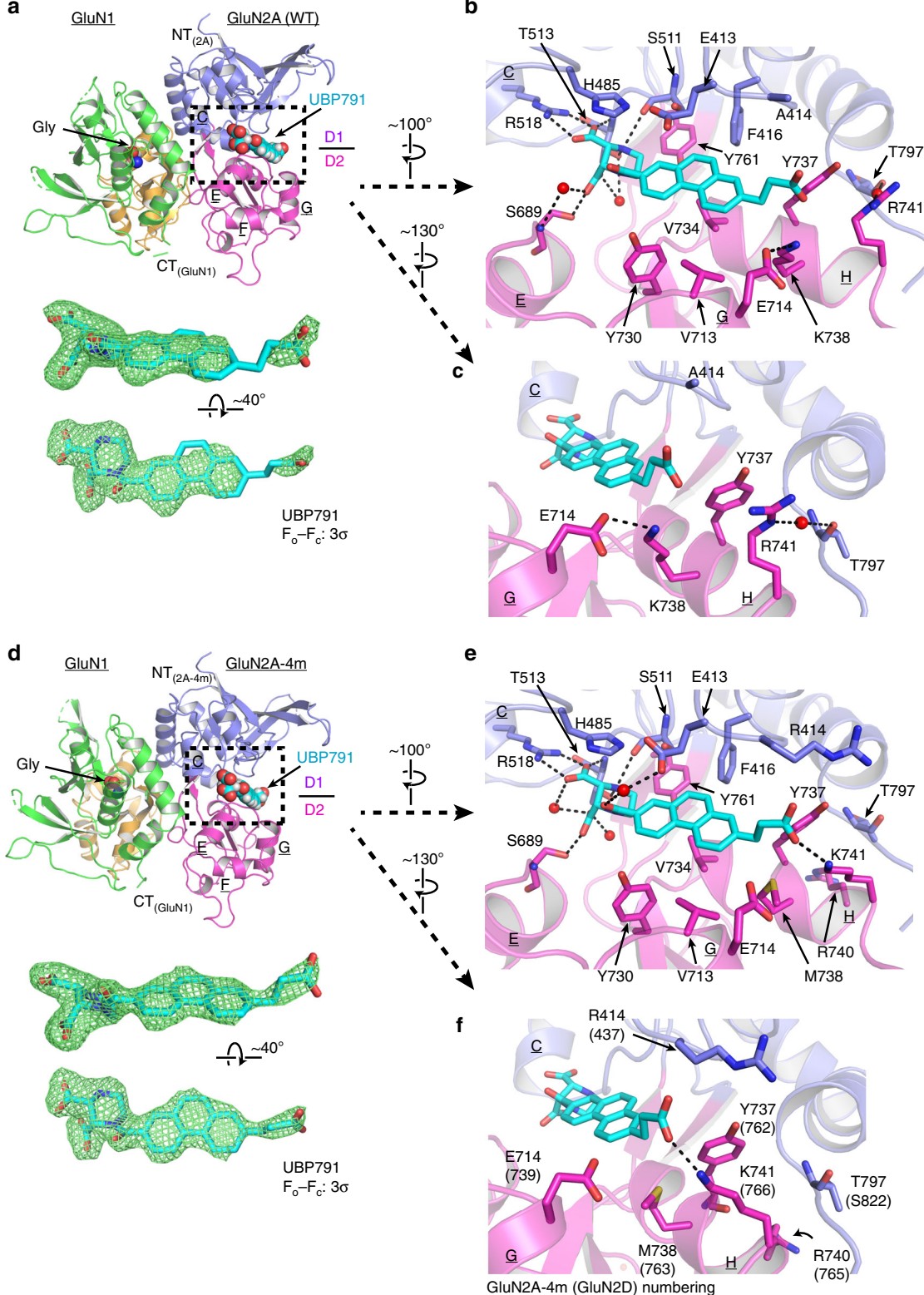

interactions involving residues GluN2A/GluN2A-4m-Thr513, Arg518, Ser511 and Ser689 whereas the phenanthrene ring forms Van der Waals interaction with GluN2A/GluN2A-4m-Tyr730, Val734, Tyr761, Val713, Phe416, Tyr737, and Ala414 in GluN2A (Arg414 in GluN2A-4m) (Fig. 5b, e). In contrast, we observed the major differences in the protein-ligand binding mode between the carboxyethyl group of UBP791 and the GluN2A LBD or the GluN2A-4m LBD (Fig. 5c, f).

In the GluN1–GluN2A-4m LBD, several elements favor accommodation of the carboxyethyl group. The most notable is the specific polar interaction with the amino group of GluN2A-4m-Lys741 (Lys766 in GluN2D) (Fig. 5e, f). This interaction is made possible by the hydrophobic interaction of GluN2A-4m-Lys741 with GluN2A-4m-Met738 (Met763 in 2D), which orients the GluN2A-4m-Lys741 side chain toward UBP791. GluN2A-4m-Met738 is positioned to form sulfur-aromatic interactions

**Fig. 5 Structures of glycine/UBP791 complexed to GluN1–GluN2A (WT) and GluN1–GluN2A-4m LBD. a** Overall structure of the GluN1–GluN2A LBD complexed to glycine (orange spheres) and UBP791 (cyan spheres) colored as in Fig. 1. Shown in mesh below is the $F_o - F_c$ omit map of UBP791 contoured at 3σ. **b**, **c** The binding site of UBP791 (cyan sticks) showing polar (dashed lines) and hydrophobic interactions. GluN2A-Lys738 and -Glu714 form a hydrogen bond whereas GluN2A-Arg741 and –Thr797 form a water-mediated hydrogen bond. **d** Overall structure of the GluN1–GluN2A-4m LBD complexed to glycine (orange spheres) and UBP791 (cyan spheres). Note that the $F_o - F_c$ omit map of UBP791 contoured at 3σ (green mesh) here is more ordered and continuous compared with that in the GluN1–GluN2A LBD in **a**. **e** The binding site of UBP791 (cyan sticks) showing similar polar (dashed lines) and hydrophobic interactions with the piperazine and phenanthrene moieties to those in the GluN1–GluN2A LBD in **a**. **f** In contrast to the GluN1–GluN2A LBD, GluN2A-4m-Met738 forms sulfur-aromatic interactions with the ligand and Tyr737, while GluN2A-4m-Lys741 forms a hydrogen bond with the carboxyethyl group of UBP791.

with the phenanthrene moiety of UBP791 as well as GluN2A-4m-Tyr737 to stabilize the binding pocket. Furthermore, GluN2A-4m-Met738 and the methylene group closest to the phenanthrene ring of UBP791 may form hydrophobic interactions. The other mutated residues GluN2A-4m-Arg414 and –Arg740 are not further involved in binding of UBP791.

The equivalent residues to GluN2A-4m-Met738 and GluN2A-4m-Lys741 in the GluN1–GluN2A LBD are GluN2A-Lys738 and GluN2A-Arg741, which are not involved in binding of UBP791. The largest difference here is that the GluN2A-Arg741 side chain faces away from the binding pocket, which is likely facilitated by charge repulsion between the amino group of GluN2A-Lys738 and the guanidinium group of GluN2A-Arg741. GluN2A-Arg741 instead forms a stacking interaction with GluN2A-Tyr737 and a water-mediated hydrogen bond with GluN2A-Thr797 (Fig. 5c). GluN2A-Lys738 is also not ideally positioned to interact with the carboxyethyl group and instead forms a polar interaction with GluN2A-Glu714 from Helix G (Fig. 5c). The unfavorable binding of UBP791 in the GluN2A binding cleft is reflected by the discontinuous and disordered electron density of the carboxyethyl group (Fig. 5a), which is in stark contrast to the density observed in GluN2A-4m (Fig. 5d). In summary, structural comparison between the GluN2A LBD and GluN2A-4m LBD (our GluN2D mimic for this study) implied that the key molecular determinants for preferential binding of UBP791 to GluN2D over GluN2A lie in the 738 and 741 positions (numbering in GluN2A) where they are lysine and arginine in GluN2A and methionine and lysine in GluN2D. Together, the methionine and lysine residues in GluN2D favorably accommodate the carboxyethyl group of UBP791 by forming both polar and hydrophobic interactions. The methionine and lysine residues are also conserved at the equivalent positions of GluN2C, thus, GluN2C-specifcity is also mediated via a similar mechanism.

**Validation of subtype-specific binding elements of UBP791.** To test the validity of the structural observation for the critical involvement of the methionine/lysine residue combination (GluN2D-Met763/Lys766, and GluN2A-4m-Met738/Lys741) in preferential binding of UBP791 to GluN2D over GluN2A, we conducted site-directed mutagenesis and assessed inhibition potencies of the mutant channels by TEVC. Specifically, we converted the GluN2D residues to the equivalent ones in the GluN2A subunit and measured macroscopic currents of the GluN1-4a/GluN2D mutant NMDARs. We first tested the single point mutants, GluN2D-Met763Lys and GluN2D Lys766Arg, which showed approximately fivefold and twofold increases in $K_i$ compared with the wildtype (WT) GluN2D, respectively (Fig. 6). The mutant GluN2A-Lys738Met (the reverse mutant of GluN2D-Met763Lys) was previously shown to increase PPDA potency by fivefold compared with the WT GluN2A[40]. Thus, our present result on GluN2D-Met763Lys strongly supported the interaction between GluN2D-Met763 and the phenanthrene backbone contained in both PPDA and UBP791. The modest change in the $K_i$ value of GluN2D Lys766Arg may be attributed to

the possibility that, in the absence of potential charge repulsion as seen in the UBP791-bound GluN2A WT LBD structure, the arginine side chain could still orient itself to form some interaction with the carboxyethyl group, hence we next tested the double mutant GluN2D-Met763Lys/Lys766Arg. In line with our structural observations, this double mutant lowered UBP791 potency by 13-fold compared with the WT GluN2D, demonstrating that GluN2D-Met763 and -Lys766 synergistically contribute to subtype-selective UBP791 binding (Fig. 6d).

**Restricted compound conformation raises subtype-selectivity.** Based on the UBP791-bound crystal structures above, we further developed three more PPDA-derivatives that aimed to improve selectivity and potency by mediating effective interactions with the critical GluN2D-specific residues around the ligand-binding pocket, GluN2D-Met763 and -Lys766. For this purpose, three compounds with the R-group containing carboxylate moieties of different length, bulkiness, and rigidity were synthesized (Fig. 7a) and tested for changes in inhibition potencies using TEVC electrophysiology. The largest change observed was for UBP1700 ((2S*,3R*)-1-(7-(2-carboxyvinyl)phenanthrene-2-carbonyl)piperazine-2,3-dicarboxylic acid), with about 50- to 60-fold GluN2C/2D-selectivity over GluN2A and 40- to 50-fold selectivity over GluN2B, which represents a substantial improvement over UBP791. While potency improved for all of the subtypes tested, the $K_i$ values for GluN2C and GluN2D are remarkably low at ~7–9 nM, making UBP1700 one of the most potent GluN2 antagonists to date (Fig. 7b, Supplementary Fig. 4). We speculate that the rigidity of the carboxyvinyl group of UBP1700 resulting from the incorporation of the double bond restricts its conformation to one that leads to favorable interactions with GluN2D-Met763 and -Lys766. The improved effect of UBP1700 was also observed on the GluN2D-mimic, GluN2A-4m, further demonstrating that the four mutations at the ligand-binding pocket can facilitate GluN2A to possess GluN2D-like ligand-binding properties (Fig. 7b). The other two compounds, UBP1701 and UBP1702, showed smaller incremental improvements with regards to selectivity (Fig. 7c). While all the compounds we tested have been cis-configured diastereomers at the 2- and 3-positions of the piperazine moiety, UBP1701 has an additional chiral center in the R-group (Fig. 7a), thus, leaving the possibility that one of the eight stereoisomers in the UBP1701 mixture could show improved selectivity.

**UBP791 and UBP1700 are highly NMDAR selective.** It has been previously shown that (2S,3R)-PPDA weakly inhibits non-NMDARs including GluA2 and GluK1 ($K_i$ of 7.85 and 1.17 μM)[40]. Similarly, other piperazine-2,3-dicarboxylic acid derivatives can inhibit both NMDAR as well as GluK1[29]. In contrast, UBP791 and UBP1700 only minimally act on non-NMDARs (Supplementary Table 2, Supplementary Fig. 5), a significant improvement particularly on GluK1 receptors ($K_i$ of UBP791~ 100 μM, of UBP1700 ~ 32 μM) (Supplementary Fig. 5). Comparisons between our UBP791-bound

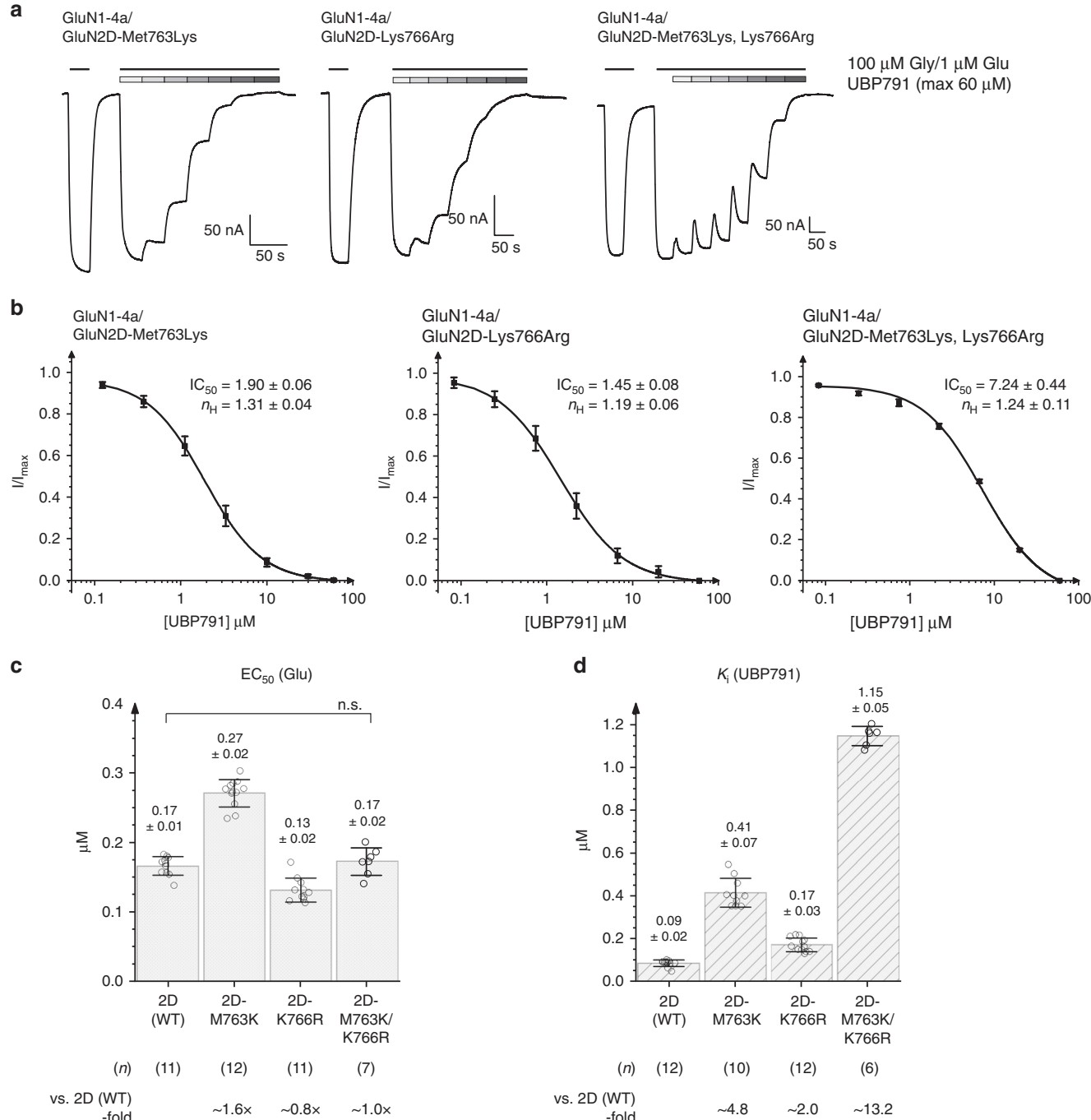

**Fig. 6 Effect of mutations in GluN2D on ʟ-glutamate and UBP791 sensitivity. a** Representative TEVC dose-response traces of single mutant GluN1-4a/ GluN2D Met763Lys (left panel), Lys766Arg (middle panel) or double mutant Met763Lys/Lys766Arg (right panel) NMDARs held at −60 mV. Currents were evoked by application of 100 μM glycine and 1 μM ʟ-glutamate and inhibited by varying concentrations of UBP791 (Met763Lys: concentration increments: 0.12/0.37/1.1/3.3/10/30/60 μM; for Lys766Arg and double mutant: three-fold increments from 0.08–60 μM). **b** Averaged dose-response curves (mean ± s.d.) for inhibition with UBP791 from eight, twelve, and six recordings for GluN1-4a/GluN2D Met763Lys, GluN1-4a/GluN2D Lys766Arg, and GluN1-4a/GluN2D Met763Lys/Lys766Arg, respectively, fit with the Hill equation to calculate $IC_{50}$ and Hill coefficient ($n_H$). **c** $EC_{50}$ for ʟ-glutamate and **d** $K_i$ for UBP791 for the mutants were obtained by TEVC recordings as in Fig. 2. Single data points are shown as open circles, the bar graph represents the mean with error bars for s.d., the number of recordings (n) and the fold-difference to $EC_{50}$ and $K_i$ of GluN2D (WT) are as shown. Pairwise comparison shows WT and mutants have different potencies ($p < 0.05$ with two-tail t-test and Bonferroni correction) except where stated (n.s.).

GluN2 LBD structures with the antagonist-bound GluA2 LBD (PDB: 1FTL)[44] or GluK1 LBD (4YMB)[45] reveal potential steric clashes between the PPDA-backbone moiety and the residues GluA2-Glu726/Met729 (GluA1-Glu719/Met722) and GluK1-Glu753 (GluK2-Glu738), as well as charge repulsion between the carboxyethyl group of UBP791 and GluK1-Glu457 (GluK2-Glu441) (Supplementary Fig. 6). Hence, insensitivity of PPDA, UBP791 and UBP1700 on AMPAR is likely due to steric clashes with piperazine/phenanthrene moieties, while increased insensitivity of UBP791 and UBP1700 on kainate receptors can likely be attributed to charge repulsion resulting from the additional carboxyethyl or carboxyvinyl group.

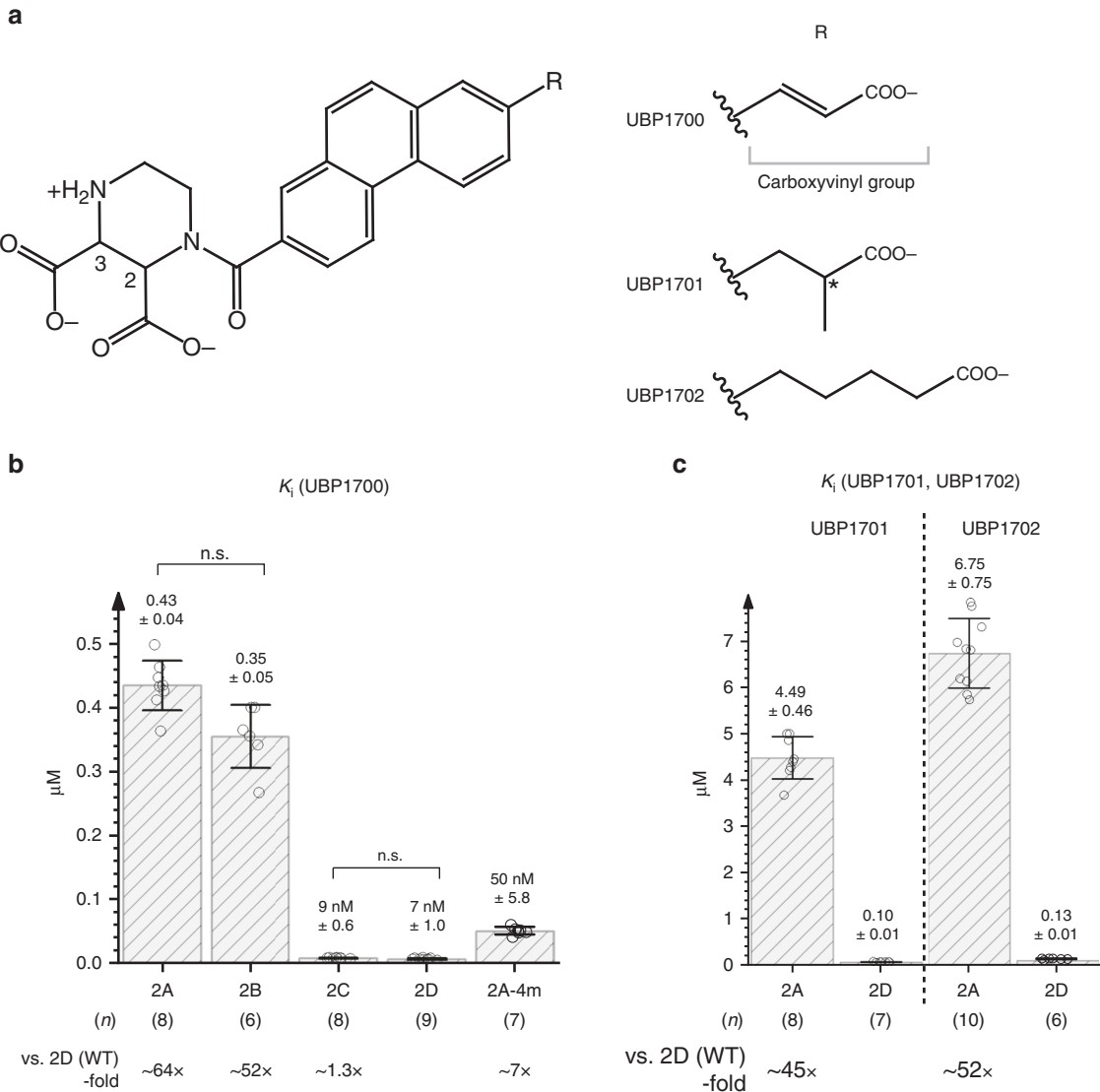

**Fig. 7 Designed compounds with increased GluN2C/2D selectivity and potency. a** UBP1700, 1701, and 1702 have carboxylate moieties of different length, bulkiness, and rigidity. The chiral center in the R-group of UBP1701 is marked with an asterisk. At physiological pH the amine group of the piperazine ring is protonated and therefore positively charged and the three carboxylic acid groups are negatively charged. **b** $K_i$ values for UBP1700 inhibition were measured on GluN1-4a/GluN2 (**a**–**d**) NMDARs and GluN1-4a/GluN2A-4m NMDAR as described in Fig. 2. Inhibition dose-response was observed in the presence of 100 μM glycine, 1 or 3 μM L-glutamate, and varying concentrations of UBP1700 (Supplementary Fig. 4). Pairwise comparison shows subtypes have different potencies ($p < 0.05$ with two-tail $t$-test and Bonferroni correction) except where stated (n.s.). **c** Potencies of UBP1701 and UBP1702 on GluN1-4a/GluN2A and GluN1-4a/GluN2D were obtained as in Fig. 2.

## Discussion

Highly potent subtype-specific reagents of NMDARs have long been heavily sought after for uses in molecular neuroscience research and for therapeutic interventions. The current study delineated the critical molecular elements for ligand-binding selectivity for the GluN2C/2D subunits over the GluN2A/2B subunits to be a set of methionine and lysine residues in the ligand-binding pocket of the GluN2C/2D LBD. UBP791, the main compound studied here, presented substantial improvement in GluN2C/2D-selectivity over GluN2A/2B (16–50×) compared with its prototype, PPDA (2.5–6×). Our crystal structures and the series of electrophysiological studies confirmed the involvement of the carboxyethyl group of UBP791 in subtype-specific binding and further led to development of UBP1700 with higher potency and subtype-selectivity. Overall, our study serves as a proof-of-principle that the LBD can be effectively targeted for subtype-specific control of NMDARs.

In this study, we used GluN2A-4m as a proxy in structural studies due to the technical difficulty in structural analysis of the GluN2D subunit. To date, the only structural work on GluN2D remains that of the GluN2D LBD complexed to agonists and partial agonists[38,39]. Nevertheless, the GluN2A-4m mutant mostly but not fully recaptures GluN2D-like functional properties including high L-glutamate potency as well as selectivity of UBP791 and UBP1700 over GluN2A/2B. The remainder of the GluN2D effect likely stems mainly from the ATD that is known to control L-glutamate potency allosterically[46,47]. Nevertheless, the structures of the agonist-bound GluN2A-4m LBD and GluN2D LBD[38,39] were similar to each other. We suggest that our antagonist-bound GluN2A-4m LBD structure would be similar to the actual antagonist-bound structure of GluN2D LBD since the binding residues for UBP791 in GluN2A-4m LBD are identical in GluN2D LBD and the specific mode of antagonist-binding mediates the domain opening of the D1 and D2 lobes. Buttressing

this prediction is the fact that our site-directed mutagenesis experiments on the GluN1/GluN2D receptors which were designed based on the UBP791-bound GluN2A-4m LBD structure showed consistent results with the structural observations (Fig. 6).

The improved subtype-selective binding of UBP791 over PPDA is attributed largely to a lowered potency for GluN2A ($K_i = \sim 0.5 \mu M$ for PPDA and $K_i = \sim 4 \mu M$ for UBP791) while maintaining potency for GluN2C/2D. The PPDA- and UBP791-bound GluN2A LBD structures did not show significant difference in the overall protein conformation suggesting that the altered potency in GluN2A may be driven by changes in ligand-binding affinity. Binding affinity can be described in thermodynamic terms as enthalpy and entropy changes where enthalpy changes reflect ligand-protein interactions and entropy changes include conformational entropy (loss of conformational degrees of freedom upon binding) and desolvation entropy (decreasing exposure of hydrophobic groups to aqueous solvent)[48]. Binding of PPDA, UBP791 and likely UBP1700 is mediated by similar sets of Van der Waals interactions, hydrogen bonds, and exclusion of solvent around the piperazine and phenanthrene moieties. Binding of UBP791 to GluN2A may result in an unfavorable conformational entropy change due to the loss of freedom around the carboxyethyl group. For UBP791 binding to GluN2C/2D, the enthalpy gains derived from the interaction between the carboxyethyl group and the methionine/lysine likely compensate for the conformational entropy loss. We can extend these considerations to a comparison between UBP791 and UBP1700. With an assumption that the methionine and lysine residues would still be able to form similar interactions with the carboxyvinyl group of UBP1700, the restricted conformation of carboxyvinyl would decrease the loss of conformational entropy and improve the affinity compared with UBP791. Such steric restriction in compounds to minimize conformational entropy loss has been a common strategy in drug development[49], though it is also known that in some cases flexible ligands favor entropic binding[50].

Given the increased interest in the physiological and pathological roles of GluN2C- and GluN2D-containing NMDARs, multiple non-competitive GluN2C/2D-selective compounds have been identified and optimized in recent years, including allosteric inhibitors QNZ46[33,51] and DQP-1105[34] with ~50-fold GluN2C/2D preference over GluN2A/2B, and NAB-14 with about 800-fold preference[36]. A current shortfall for these compounds is their low potency with $IC_{50}$ values in the micromolar range for GluN2C/2D. While their binding modes have not been well defined due to the lack of structures, mutagenesis studies on GluN2D predicted the structural determinants for inhibition of QNZ46 and DQP-1105 to lie in the GluN2D LBD-S2 lobe, close to the linker that connects the LBD to the TMD[33,34,51], whereas those of NAB-14 lie in the M1 helix region of the GluN2D subunits[36]. The UBP compounds studied here display similar GluN2C/2D selectivity as QNZ46 and DQP-1105, but lower selectivity compared with NAB-14. However, the UBP compounds possess higher potencies ($K_i$ in the low nanomolar range), thus likely inhibiting the receptors more effectively than these allosteric compounds in environments with lower agonist concentrations such as peri- and extrasynaptic spaces where GluN2D-containing NMDARs are found. Furthermore, we expect the UBP compounds to have similar inhibition potencies on tri-heteromeric NMDARs (e.g. GluN1/GluN2B/GluN2D) as on di-heteromeric NMDARs (e.g. GluN1/GluN2D). This is because opening of NMDAR channels requires Gly and glutamate binding to all of the four subunits, thus, binding of the UBP compounds to even one GluN2 subunit would be sufficient for inhibition. With respect to application in animal studies, UBP791 and UBP1700 have not

been used in vivo yet, but previous studies with PPDA-derivatives suggest that the UBP compounds might be able to cross a healthy blood-brain barrier[29], and can cross a compromised blood-brain barrier[52].

While the potencies of the UBP compounds are high, it would be desirable to improve their GluN2C/2D-selectivity. The present series of the UBP compounds interact with the methionine and lysine residues unique to GluN2C/2D, but not with the third unique residue, GluN2D-Arg437 (GluN2C-Arg411), which is in spatial proximity to the UBP compounds. Thus, we predict that the UBP compounds that could additionally engage GluN2D-Arg437 (GluN2C-Arg411) in ligand binding will display improved selectivity for GluN2C/2D-containing NMDARs. Ultimately, it would be ideal to develop a compound that can distinguish GluN2C and GluN2D. One possibility for achieving this may be to target GluN2D-Pro736/Arg737 or the equivalent GluN2C-Arg709/Ser710. In addition, in all of the cases, the selectivity and potency will likely improve by purifying and isolating specific enantiomers. Nevertheless, the current study demonstrates that the chemical nature of the ligand-binding pocket of the LBD can be exploited with the piperazine-phenanthrene backbone and opens the avenue for further improvements in the specific targeting of GluN2C- or GluN2D-containing NMDARs.

## Methods

**Compound synthesis**. Compound synthesis and reaction schemes are included in the Supplementary Methods.

**Electrophysiology**. Plasmids (pSP or pCI_NEO) harboring rat GluN1 or GluN2 full-length subunits were linearized by restriction digestion and transcribed according to the manufacturer's instructions (mMESSAGE mMachine SP6 or T7 kit, ThermoFisher Scientific). Recombinant rat GluN1-4a and GluN2 were expressed by co-injecting cRNA at a 1:2 (w/w) ratio into defollicated X. laevis oocytes (amounts varied from 0.05–25 ng per oocyte). Two-electrode voltage-clamp recordings were performed 1–3 days after injection using agarose-tipped microelectrodes ($0.4$–$1.2 M\Omega$) filled with 3 M KCl at a holding potential of $-60$ mV. The bath solution contained 5 mM HEPES, 100 mM NaCl, 0.3 mM $BaCl_2$, 10 mM Tricine, at pH 7.4 (adjusted with KOH). Currents were evoked by application of 100 μM Gly and various concentrations of L-glutamate, with and without addition of inhibitors at various concentrations. For solubility, DMSO (final 0.1–0.5%) was added to solutions containing UBP compounds, and bath solutions were matched accordingly. Data were acquired and analyzed using the software program Pulse (HEKA, Holliston, MA), fitted using the program IGOR to obtain $EC_{50}$ values of L-glutamate and $IC_{50}$ values of antagonists. $K_i$ values were subsequently calculated using the Cheng–Prusoff equation $K_i = IC_{50}/(1 + [L\text{-glu-}tamate]/EC_{50})$. For the statistical evaluation, we implemented pairwise $F$-tests to test variances, and then performed the Student $t$ test with equal or unequal variances as determined previously, and accounted for multiple testing with the Bonferroni correction.

**Expression and purification of GluN1 and GluN2A LBD**. Plasmid constructs, protein expression and purification generally followed previously published protocols[40,41]. All of the DNAs encoding NMDARs used in this study were from Rattus norvegicus (rat). GluN2A-4m (mutant) differed from the WT GluN2A sequence in the following four residues: Ala414Arg, Lys738Met, Gly740Arg, Arg741Lys. GluN1 LBD was composed of Met394-Lys544 (GluN1-S1) and Arg663-Ser800 (GluN1-S2), connected by a Gly-Thr dipeptide linker (Fig. 1b). GluN2A or GluN2A-4m was composed of Asp402-Arg539 (GluN2A-S1) and Gln661-Asn802 (GluN2A-S2), connected by a Gly-Thr dipeptide linker. The GluN1 LBD construct was fused with an octa-Histidine ($His_8$) tag followed by a thrombin cleavage site at the N-terminus whereas the GluN2A and GluN2A-4m LBD constructs were fused to $His_6$-SUMO at the N-termini. OrigamiB (DE3) Escherichia coli cells (Novagen) harboring the LBD constructs in pET22b(+) without the pelB sequence were grown at 37 °C to $OD_{600}$ of 1.5 and induced with 0.5 mM Isopropyl β-D-1-thiogalactopyranoside (IPTG) at 15 °C for protein expression.

GluN1 LBD was purified by Ni-nitrilotriacetic acid (NTA) affinity chromatography, digested with thrombin to remove the $His_8$-tag, and further purified by SP-Sepharose cation exchange chromatography (GE Healthcare), all in the presence of 1 mM Gly. GluN2A or GluN2A-4m LBD was purified by Ni-NTA affinity chromatography, digested with ubiquitin ligase protease-1 to remove the $His_6$-SUMO-tag, and then further purified by Q-Sepharose anion exchange chromatography (GE Healthcare) and SP-Sepharose cation exchange chromatography, all in the presence of 1 mM L-glutamate (Glu).

**Crystallization and soaking**. The purified GluN1 and GluN2A or GluN1 and GluN2A-4m LBD protein were separately concentrated to about 6 mg ml$^{-1}$, mixed in a 1:1 weight ratio, and dialyzed against 10 mM HEPES (pH 7.0), 100 mM NaCl, 1 mM Gly, and 1 mM Glu. Crystallization was conducted using hanging-drop vapor diffusion at 18 °C where reservoir solutions contained 100 mM HEPES (pH 7.0), 60–90 mM NaCl, and 15–20% polyethylene glycol monomethylether 2000 (PEG2000 MME). Drops were mixed at 2:1 and 3:1 (protein:reservoir) volume ratios. Gly/Glu-bound GluN1–GluN2A-4m crystals were briefly soaked against a buffer with their reservoir solution supplemented with 1 mM Gly, 1 mM Glu and 18% glycerol for a few seconds and flash frozen. For Gly/HQA-bound crystals of GluN1–GluN2A LBD and GluN1–GluN2A-4m LBD, the pooled protein samples were dialyzed first against 10 mM HEPES (pH 7.0), 100 mM NaCl, and 1 mM Gly and later against the same buffer containing 100–200 μM HQA. Prior to setting up hanging drops for crystallization, 2–5 mM HQA was added to the protein mixture, which was then equilibrated for 5 min, and centrifuged at $20,000 \times g$ for 10 min. Gly/HQA-bound GluN1–GluN2A LBD or GluN1–GluN2A-4m LBD crystals were briefly soaked against a buffer with their reservoir solution supplemented with 1 mM Gly, 5 mM HQA and 18% glycerol for a few seconds and flash frozen.

For Gly/UBP791-bound GluN1–GluN2A LBD, the GluN1–GluN2A LBD crystals formed in the presence of Gly/Glu were briefly 'washed' three times in the reservoir solution and soaked in new drops containing 100 mM HEPES (pH 7.0), 18–21% PEG2000 MME, 75 mM NaCl, 1 mM Gly, and 300 μM UBP791 for 24 h twice. The crystals were soaked in new drops supplemented with up to 10 mM UBP791 for 12–24 h, subsequently soaked against the same buffer containing 18% glycerol for a few seconds and flash frozen in liquid nitrogen. The same procedure was used to obtain the Gly/UBP791-bound GluN1–GluN2A-4m LBD crystals except that the Gly/HQA-bound GluN1–GluN2A-4m LBD crystals were used for soaks against UBP791.

**Structural analysis**. X-ray diffraction data were collected at the ID-17 beamline of the National Synchrotron Light Source II at Brookhaven National Laboratory at the wavelength of 0.920 Å and at 100 K and processed using HKL2000[53], FastDP[54], or the autoPROC suite[55], which include XDS[56], POINTLESS[57], AIMLESS[58], CCP4[59], and STARANISO[60]. All of the structures were determined by molecular replacement using the PDB coordinates 4NF8 and 4NF6 as search probes. Molecular replacement, structural refinement and model building were performed using PHASER[61], PHENIX[62], and Coot[63]. All of the refined structural models had more than 95% favored geometry according to Ramachandran statistics. The ligands were built in ChemDraw and CIF files were generated in the Grade server (Global Phasing Ltd). Extent of the D1–D2 domain opening and closing was calculated by superposing the Cαs of the D1 residues then calculating rotational angles required to superpose the Cαs of the D2 residues. The rotational angles were calculated using the PyMOL script draw_rotation_axis.py (https://pymolwiki.org/index.php/RotationAxis).

**Reporting summary**. Further information on research design is available in the Nature Research Reporting Summary linked to this article.

## Data availability

Data supporting the findings of this manuscript are available from the corresponding authors upon reasonable request. A reporting summary for this Article is available as a Supplementary Information file. The source data underlying Figs. 2c, d, 3, 4c–e, Supplementary Figs. 2, 4, and 5, and Supplementary Table 2 are provided as a Source Data file. Atomic coordinates and structure factors for the GluN1/GluN2A LBDs with glycine and homoquinolinic acid, and with glycine and UBP791 are deposited to the Protein Data Bank under the accession codes, 6UZR and 6UZW, respectively. The coordinates for GluN1/GluN2A-4m LBDs with glycine and glutamate, with glycine and homoquinolinic acid, and with glycine and UBP791 are deposited under the accession codes 6UZ6, 6UZG, and 6UZX, respectively.

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

## Acknowledgements

We thank the staff at the 17-ID beamlines at the Brookhaven National Laboratory NSLS-II, in particular Jean Jakoncic, Alexei Soares and Vivian Stojanoff, for help during data collection. This work was funded by NIH (NS111745 and MH085926), Robertson funds at Cold Spring Harbor Laboratory, Doug Fox Alzheimer's fund, Austin's purpose, Heartfelt Wing Alzheimer's fund (to H.F.), NIH (MH060252) (to D.T.M. and D.E.J.), BBSRC (BB/L001977/1) (to D.E.J.) and European Research Council (to G.L.C.). J.X.W. was supported by the George A. & Marjorie H. Anderson Fellowship and a PhD fellowship of the Boehringer Ingelheim Fonds.

## Author contributions

J.X.W. and H.F. designed and conducted experiments involving X-ray crystallography and electrophysiology. M.L. conducted electrophysiology. N.S. conducted molecular biology for all aspects of the experiments. M.W.I., E.S.B., K.S., R.J.T., A.V., G.L.C., D.T.M. and D.E.J. were involved in synthesis and characterization of the UBP compounds by the calcium flux assay. J.X.W., H.F., M.W.I., R.J.T., D.E.J. wrote the manuscript.

## Competing interests

The authors declare no competing interests.
