## [Peer Review File · Nature Communications]

Reviewers' Comments:

Reviewer #1:

Remarks to the Author:

Ligands selective for GluN2A and GluN2B subunit containing NMDA receptors are already available. However, ligands for GluN2C and GluN2D subunit containing NMDA receptors are very limited. In this manuscript, electrophysiology, site directed mutagenesis and X-ray crystal structure were used to analyze the glutamate binding site of NMDA receptors containing GluN2C or GluN2D subunits. At first, interaction of phenanthrene derivative UBP791, which is derived from 1-(phenanthrylcarbonyl)piperazinedicarboxylic acid (PPDA) by addition of a carboxyethyl moiety, with NMDA receptor subtypes was analyzed. UBP791 shows increased selectivity for GluN2C/2D receptors over GluN2A/2B receptors. Four amino acids in the ligand-binding domain of GluN2A and GluN2D differ. These amino acids are Ala414, Lys738, Gly740 and Arg741 in the GluN2A subunit. These four amino acids were mutated into the corresponding amino acids of the GluN2D subunit, which resulted in a ligand-binding domain with similar properties as the GluN2D LBD, i.e. the GluN2A-4m LBD serves as structural proxy for the GluN2D LBD. The X-ray crystal structure of GluN2A-LBD and GluN2A-4m LBD with UBP791 was recorded and analyzed in detail leading to an explanation of the increased subtype-selectivity of UBP791 towards the GluN2A-4m LBD. In particular, Met763 and Lys766 in GluN2D LBD instead of Leu738 and Arg741 in GluN2A LBD were identified as selectivity determining structural elements. These results were exploited by the development of UBP1700 bearing a carboxyvinyl moiety at the phenanthrene ring system instead of the carboxyethyl moiety of UBP791. Due to the conformationally restricted side chain, UBP1700 shows low nanomolar potency at GluN2C/2D receptors as well as more than 40-fold selectivity over GluN2A and GluN2B.

Exceptional performance of this manuscript: (1) The structural analysis of GluN2D-LBD with agonists and partial agonists has been performed, however, crystal structure analysis of GluN2D-LBD with agonists was not possible so far. Here, the 4-fold mutant of the GluN2A-LBD, i.e. GluN2A-4m-LBD, was used as proxy of the GluN2D-LBD. The GluN2A-4m could be crystallized with the antagonist UBP971 providing structural information of a bound antagonist to the GluN2D binding pocket for the first time. (2) Compared to allosteric GluN2C/2D inhibitors the new ligand UBP1700 shows considerably increased potency at GluN2C/2D receptors, i.e. for the first time, a GluN2C/2D-selective and potent ligand is available to study the properties of these NMDA receptor subtypes. Publication of these excellent results in Nature Communications is strongly recommended.

Minor comments

1. Page 16, line 9: "cis-configured diastereomers at the 2- and 3-position" instead of "cis-enantiomers".
2. Page 16, line 11: "one of the eight stereoisomers" instead of "four enantiomers". UBP1701 has three centers of chirality, which results in 8 stereoisomers or four diastereomeric pairs of enantiomers.

Reviewer #2:

Remarks to the Author:

Wang et al. report the compound UBP791, which exhibits 47-fold and 16-fold binding preference for GluN2C/2D-containing NMDARs over GluN2A/2B-containing ones. A methionine and lysine unique to GluN2C/2D was found to confer selective binding of UBP791, determined using X-ray crystallography and electrophysiology. Because antagonist-bound GluN2D LBDs have proven technically challenging for structural work, the authors used a GluN2D-like structural proxy, GluN2A-4m, which differs from WT GluN2A by four residues. Structure-based modification of UBP791 led to UBP1700, an improved compound that exhibited 63- and 52-fold selectivity for GluN2D, and 50- and 40-fold selectivity for GluN2C, over GluN2A and GluN2B. This study demonstrates that despite high sequence conservation among GluN2 LBDs, their agonist-binding

pockets are potential subtype-specific targets for small compounds.

The manuscript is clear and well written, and the experiments are logical and appropriately analyzed. The work is an important step forward in structure-based design of compounds that bind specific NMDA receptor subtypes.

1. Figure 1c shows the structures of the GluN2 agonists and antagonists. However, at the pH values at which the experiments were carried out (pH 7.4 for electrophysiology, pH 7 for crystallization), it is unlikely that these compounds would adopt the protonation states as drawn. Indicating the dominant form of the compound at the experimentally relevant pH would aid in the interpretation of the crystal structures (e.g., a negative carboxyethyl group would experience greater stabilization by the critical lysine residue than a protonated carboxyethyl).

2. If both enantiomers of UBP791 do bind, the observed functional effect would be a combination of both (2S, 3R) and (2R, 3S). In this case, the less potent enantiomer could potentially compete with the more potent one, resulting in decreased overall potency than would be observed with the isolated (2S, 3R)-UBP791. It might be useful to discuss this possibility.

3. Last line of page 12: is (2S,3R)-PPDA 10-fold more potent than (2R,3S)-PPDA for GluN2D or both GluN2D/2C?

4. Was the degree of domain opening calculated using Hingefind? I did not see this stated in the manuscript.

5. In Fig. S3, please change the yellow labels (add outline, add background, darken, etc.) to improve legibility.

Reviewer #3:

Remarks to the Author:

The manuscript is well written and clearly presented. Subtype-specific pharmacological targeting of NMDA receptors is currently extensively studied as a valuable research tool and for its great potential in the treatment of a variety of human diseases. Authors used combination of TEVC electrophysiology and X-ray crystallography to identify important amino acid residues in the glutamate binding pocket of GluN2D subunit that are critical for subtype-specific binding of competitive inhibitors based on piperazine/phenanthrene core structure. This incrementally led to the development of compound UBP1700 which shows ~40 to 60-fold GluN2C/2D-selectivity over GluN2A-2B with affinity in the low nanomolar range which is about 5 to 15-fold improvement in selectivity and ~200-fold increase in affinity over the already published piperazine/phenanthrene analogs.

1) As the authors mentioned in the manuscript that Swanger A. et al (2007) characterized a novel NMDA negative allosteric modulator NAB-14 that is >800-fold selective for recombinant GluN2C/GluN2D over GluN2A/GluN2B and has an IC₅₀ value of 580 nM at recombinant GluN2D-containing receptors. Authors should explain how the UBP1700 has advantages over NAB-14. For example, UBP1700 has higher affinity over NAB-14. Would the high affinity have an advantage in clinical use? Can authors provide any prediction about the Blood-brain barrier permeability of UBP1700 or toxicity?

2) It was shown that NMDA receptors are expressed in mammalian CNS not only as di-heterotetramers but also as tri-heterotetrameric receptors that are composed of two different GluN2 subunits. Can authors provide some experimental data showing the effect of UBP1700 on these receptors or at least comment on what the expected outcome of this drug on them going to be? Is there any expected advantage of using competitive inhibitors on glutamate binding site over

negative allosteric modulators considering tri-heteromeric receptors and dynamic nature of glutamate neurotransmission in CNS?

3) To predict the inhibitory outcome of competitive inhibitors on synaptic NMDA receptors it is important to know their binding and unbinding rate constants. Can authors provide these rate constants?

4) Lind GE, et al. (2017) published crystal structures of the GluN1/2A agonist binding domain heterodimer with bound ACEPC antagonists that reveal a binding mode in which the ligands occupy a cavity that extends toward the subunit interface between GluN1 and GluN2A ABDs. Can authors speculate whether this cavity could be used to further improve specificity and/or affinity of UBP1700?

5) Why did the authors use different assays to determine drug affinity for different ionotropic glutamate receptors? TEVC for GluN1/GluN2A-D and glutamate-stimulated Ca²⁺-influx assay for GluA1, GluK1, and GluK2?

6) Authors should comment on why racemic mixtures of drugs instead of pure enantiomers were used.

7) Authors couldn't crystalize GluN2D competitive antagonist bound LBD, so x-ray results are done on GluN2A-4m construct, which is GluN2A ligand-binding domain that mimics the GluN2D structure by introducing four key amino acid mutations into the GluN2A subunit. Authors provide evidence that in agonist bound state GluN2A-4m and GluN2D show very similar structure, however, it is not clear to what extent is this also valid for antagonist bound state. This should be addressed.

8) Authors should add hill coefficient values into the dose-response plots.

Minor points:

1) Citation format on page 5 should be changed "(Furukawa & Gouaux, 2003; Furukawa et al, 2005; Karakas et al, 2011)"

2) glutamate, L-glutamate, L-Glutamate, L-glutamate; and Parkinson's disease vs Parkinson's Disease should be named consistently

3) In figure 3d, "(12)" shouldn't be inside of the plot

Dear Editors & Editorial Staff,

We recently received several comments on our submitted manuscript, “**Structural basis of subtype-selective competitive antagonism for GluN2C/2D-containing NMDA receptors**” (Ref. No. NCOMMS- 19-33444). We wish to thank all of the reviewers and Nature Communications staff for their thoughtful comments, which we have addressed below. For clarity, we directly respond to each comment in turn, with the reviewers’ comments in italic.

Reviewers' comments:

Reviewer #1 (Remarks to the Author):

Ligands selective for GluN2A and GluN2B subunit containing NMDA receptors are already available. However, ligands for GluN2C and GluN2D subunit containing NMDA receptors are very limited. In this manuscript, electrophysiology, site directed mutagenesis and X-ray crystal structure were used to analyze the glutamate binding site of NMDA receptors containing GluN2C or GluN2D subunits. At first, interaction of phenanthrene derivative UBP791, which is derived from 1(phenanthrylcarbonyl)piperazinedicarboxylic acid (PPDA) by addition of a carboxyethyl moiety, with NMDA receptor subtypes was analyzed. UBP791 shows increased selectivity for GluN2C/2D receptors over GluN2A/2B receptors. Four amino acids in the ligand-binding domain of GluN2A and GluN2D differ. These amino acids are Ala414, Lys738, Gly740 and Arg741 in the GluN2A subunit. These four amino acids were mutated into the corresponding amino acids of the GluN2D subunit, which resulted in a ligand-binding domain with similar properties as the GluN2D LBD, i.e. the GluN2A-4m LBD serves as structural proxy for the GluN2D LBD. The X-ray crystal structure of GluN2A-LBD and GluN2A-4m LBD with UBP791 was recorded and analyzed in detail leading to an explanation of the increased subtype-selectivity of UBP791 towards the GluN2A-4m LBD. In particular, Met763 and Lys766 in GluN2D LBD instead of Leu738 and Arg741 in GluN2A LBD were identified as selectivity determining structural elements. These results were exploited by the development of UBP1700 bearing a carboxyvinyl moiety at the phenanthrene ring system instead of the carboxyethyl moiety of UBP791. Due to the conformationally restricted side chain, UBP1700 shows low nanomolar potency at GluN2C/2D receptors as well as more than 40-fold selectivity over GluN2A and GluN2B. Exceptional performance of this manuscript: (1) The structural analysis of GluN2D-LBD with agonists and partial agonists has been performed, however, crystal structure analysis of GluN2D-LBD with agonists was not possible so far. Here, the 4-fold mutant of the GluN2A-LBD, i.e. GluN2A-4m-LBD, was used as proxy of the GluN2D-LBD. The GluN2A-4m could be crystallized with the antagonist UBP791 providing structural information of a bound antagonist to the GluN2D binding pocket for the first time. (2) Compared to allosteric GluN2C/2D inhibitors the new ligand UBP1700 shows considerably increased potency at GluN2C/2D receptors, i.e. for the first time, a GluN2C/2D-selective and potent ligand is available to study the properties of these NMDA receptor subtypes. Publication of these excellent results in Nature Communications is strongly recommended.

Minor comments

- 1. Page 16, line 9: “cis-configured diastereomers at the 2- and 3-position” instead of “cis-enantiomers”.*
- 2. Page 16, line 11: “one of the eight stereoisomers” instead of “four enantiomers”. UBP1701 has three centers of chirality, which results in 8 stereoisomers or four diastereomeric pairs of enantiomers.*

Authors’ response: We agree with the reviewer - these changes were made in the revised manuscript.

Reviewer #2 (Remarks to the Author):

Wang et al. report the compound UBP791, which exhibits 47-fold and 16-fold binding preference for GluN2C/2D-containing NMDARs over GluN2A/2B-containing ones. A methionine and lysine unique to GluN2C/2D was found to confer selective binding of UBP791, determined using X-ray crystallography and electrophysiology. Because antagonist-bound GluN2D LBDs have proven technically challenging for structural work, the authors used a GluN2D-like structural proxy, GluN2A-4m, which differs from WT GluN2A by four residues. Structure-based modification of UBP791 led to UBP1700, an improved compound that exhibited 63- and 52-fold selectivity for GluN2D, and 50- and 40-fold selectivity for GluN2C, over GluN2A and GluN2B. This study demonstrates that despite high sequence conservation among GluN2 LBDs, their agonist-binding pockets are potential subtype-specific targets for small compounds. The manuscript is clear and well written, and the experiments are logical and appropriately analyzed. The work is an important step forward in structure-based design of compounds that bind specific NMDA receptor subtypes.

1. Figure 1c shows the structures of the GluN2 agonists and antagonists. However, at the pH values at which the experiments were carried out (pH 7.4 for electrophysiology, pH 7 for crystallization), it is unlikely that these compounds would adopt the protonation states as drawn. Indicating the dominant form of the compound at the experimentally relevant pH would aid in the interpretation of the crystal structures (e.g., a negative carboxyethyl group would experience greater stabilization by the critical lysine residue than a protonated carboxyethyl).

Authors' response: Unless the pH is below 4 all carboxylic acids will be negatively charged - as suggested by this reviewer. The nitrogen at the 4-position of the UBP compounds will be positively charged e.g.

Figure 1 and 7 were revised accordingly.

2. If both enantiomers of UBP791 do bind, the observed functional effect would be a combination of both (2S, 3R) and (2R, 3S). In this case, the less potent enantiomer could potentially compete with the more potent one, resulting in decreased overall potency than would be observed with the isolated (2S, 3R)-UBP791. It might be useful to discuss this possibility.

3. Last line of page 12: is (2S,3R)-PPDA 10-fold more potent than (2R,3S)-PPDA for GluN2D or both GluN2D/2C?

Authors' response: The (2S,3R) isomer of UBP791 likely has higher potency for GluN2C and GluN2D than the (2R,3S)-isomer, considering the potency differences observed for the (2S,3R) and (2R,3S) isomers of PPDA.

The reported exact values for the PPDA isomers have differed though: Irvine *et al* showed that the (-)-PPDA (which is the (2S, 3R)-isomer) has ~50-fold higher potency for GluN2C and GluN2D-containing NMDARs (see compound 4 - Irvine et al, 2012) compared to the (+)-isomer, while Jespersen *et al* showed only 7-fold higher potency for GluN2D-containing NMDARs (Jespersen et al, 2014). The discrepancy could be due to differences in the purity of the two PPDA isomers and/or differences in the recording conditions in the electrophysiological assay on GluN2 subtypes expressed in *Xenopus* oocytes.

Overall, the more potent (-)-isomer showed ~1.4-fold higher potency than the racemate on GluN2C and GluN2D (Irvine et al., 2012), highlighting that the less active isomer does not decrease the overall potency by much. This could be explained that in the lower concentration range, only the more potent – possibly higher affinity – compound will effectively bind, minimizing competition. This point is addressed in the section, “*Structures reveal molecular determinants for subtype-selective binding of UBP791.*”

4. Was the degree of domain opening calculated using Hingefind? I did not see this stated in the manuscript.

Authors' response: It was calculated by superposing D1 (the upper lobe) of the LBDs at first followed by superposing D2 (the lower lobe). The rotational angle to superpose the D2 was calculated by a publicly available pymol script, draw_rotation_axis.py (<https://pymolwiki.org/index.php/RotationAxis>). This information is included in the method section of the revised manuscript.

5. In Fig. S3, please change the yellow labels (add outline, add background, darken, etc.) to improve legibility.

Authors' response: The labelling in Fig. S3 was changed to improve the legibility. Thank you for the suggestion.

Reviewer #3 (Remarks to the Author):

The manuscript is well written and clearly presented. Subtype-specific pharmacological targeting of NMDA receptors is currently extensively studied as a valuable research tool and for its great potential in the treatment of a variety of human diseases. Authors used combination of TEVC electrophysiology and X-ray crystallography to identify important amino acid residues in the glutamate binding pocket of GluN2D subunit that are critical for subtype-specific binding of competitive inhibitors based on piperazine/phenanthrene core structure. This incrementally led to the development of compound UBP1700 which shows ~40 to 60-fold GluN2C/2D-selectivity over GluN2A-2B with affinity in the low nanomolar range which is about 5 to 15-fold improvement in selectivity and ~200-fold increase in affinity over the already published piperazine/phenanthrene analogs.

1) As the authors mentioned in the manuscript that Swanger A. et al (2007) characterized a novel NMDA

negative allosteric modulator NAB-14 that is >800-fold selective for recombinant GluN2C/GluN2D over GluN2A/GluN2B and has an IC₅₀ value of 580 nM at recombinant GluN2D-containing receptors. Authors should explain how the UBP1700 has advantages over NAB-14. For example, UBP1700 has higher affinity over NAB-14. Would the high affinity have an advantage in clinical use? Can authors provide any prediction about the Blood–brain barrier permeability of UBP1700 or toxicity?

Authors' response: The NAB-14 IC₅₀ value of 584 nM on GluN2D quoted by reviewer #3 came from an assay of GluN2D expressed in mammalian cells. In an electrophysiological assay on GluN2D expressed in *Xenopus* oocytes NAB-14 was reported to have an IC₅₀ value of 2.2 μM (Swanger et al., 2017). UBP1700 in the same oocyte-based assay had a K_i value of 7 nM, suggesting much higher potency compared to NAB-14. We realize that there is an inherent difference between allosteric compounds' IC₅₀ vs competitive antagonists' K_i values, and during signal transmission when an estimated 1 mM burst of glutamate can be briefly present in the synaptic cleft, those potency differences might be negligible, or even favor the use of the allosteric inhibitors. However, the competitive UBP1700 might be more relevant and potent in circumstances with lower glutamate concentrations, e.g. inhibit peri- and extrasynaptic NMDARs (which have been suggested to contain GluN2D), or in use with tri-heteromeric receptors.

Multiple allosteric modulators show weaker / incomplete inhibition of triheteromeric receptors (e.g. ifenprodil, TCN-201 – see Stroebel et al, 2014; Hansen et al, 2014). Similarly, using the oocyte-based system, NAB-14 was shown to inhibit GluN2A/2C-triheteromers less effectively (IC₅₀ ~15 μM, maximally inhibit to 19.9% of the maximal response), compared to the GluN2C-diheteromers (IC₅₀ 5 μM, maximally inhibited to 11.7%) (Swanger et al, 2017). – As discussed in the next question, competitive antagonist might allow full inhibition in contrast to allosteric modulators.

UBP1700 has not been used *in vivo* and so we cannot comment on its blood brain barrier penetration or toxicity. However, two analogues of UBP1700 have been tested *in vivo*. UBP145 given i.v. blocked neurotoxicity of co-applied tissue plasminogen activator (tPA) in a thrombotic stroke model in mice and prevented the deleterious effect of late thrombolysis by tPA. However, UBP145 was only active *in vivo* in the thrombotic stroke model where the blood brain barrier was compromised (Jullienne et al, 2011). UBP161 (compound 18i in Irvine et al, 2012) when given at a dose of 30 mg/kg i.p. showed antinociceptive activity in a model of mild nerve injury (Irvine et al, 2012). Given that UBP1700 has > 140-fold higher affinity on GluN2D compared to UBP145 and UBP161, it is possible that it would be able to cross the blood-brain barrier under specific circumstances and be at a concentration in the brain sufficient to block GluN2D.

In terms of clinical use, antagonists with higher affinity have the potential advantage that lower concentrations would be needed and hence there is less likelihood of side-effects. But, as is always the case, this can only be ascertained for certain with clinical trials. However, it should be noted that UBP1700 isn't being developed as a therapeutic agent. The knowledge gained from its development and use should, however, be of value to drug development programs.

2) *It was shown that NMDA receptors are expressed in mammalian CNS not only as di-heterotetramers but also as tri-heterotetrameric receptors that are composed of two different GluN2 subunits. Can*

authors provide some experimental data showing the effect of UBP1700 on these receptors or at least comment on what the expected outcome of this drug on them going to be? Is there any expected advantage of using competitive inhibitors on glutamate binding site over negative allosteric modulators considering tri-heteromeric receptors and dynamic nature of glutamate neurotransmission in CNS?

UBP1700 has not been tested on recombinant NMDAR triheteromers expressed in *Xenopus* oocytes. We would expect that UBP1700 would block triheteromers to a similar extent to that of diheteromers containing either GluN2C or GluN2D, given that it is a competitive antagonist. Opening of the NMDAR channel requires glycine to bind to each of the ligand binding domains of GluN1 subunits and glutamate to bind to each of the GluN2 subunits in a tetramer. Thus, UBP1700 most likely only needs to bind to the ligand binding domain of one GluN2 subunit in the NMDAR to block channel opening.

Analogues of UBP1700 have been shown to block triheteromers, suggesting that UBP1700 would behave similarly. UBP145 blocks what are most probably GluN1/GluN2B/GluN2D triheteromers in the hippocampus at concentrations that are selective for GluN2D (Volianskis et al., 2013). UBP141 inhibits what are thought to be GluN1/GluN2B/GluN2D triheteromers expressed in the substantia nigra pars compacta (Brothwell et al., 2008; Suárez et al., 2010).

Negative allosteric modulators (NAMs) based on ifenprodil such as Ro 25-6981 have very different responses on diheteromeric GluN1/GluN2B compared to triheteromeric GluN1/GluN2A/GluN2B, such that it has much greater antagonist potency on diheteromeric receptors. NAB-14 was also shown to have lower antagonist potency on triheteromeric GluN1/GluN2A/GluN2C than on diheteromeric GluN1/GluN2C (Swanger A, et al., 2007).

Thus, competitive antagonists may have an advantage in being able to block both diheteromeric and triheteromeric combinations. However, NAMs may have greater utility, as it may be possible to design them to be selective for either diheteromeric or triheteromeric NMDARs. Ultimately, both competitive antagonists and NAMs will need to be evaluated in animal models of neurological disorders and in human clinical trials to discover which are best suited towards therapeutic applications. A part of the above argument is included in the revised manuscript.

3) To predict the inhibitory outcome of competitive inhibitors on synaptic NMDA receptors it is important to know their binding and unbinding rate constants. Can authors provide these rate constants?

Authors' response: We have not measured binding and unbinding rate constants yet. Once more compounds with improved selectivity/potency are developed based on UBP1700, we plan to measure inhibition kinetics using patch-clamp electrophysiology with fast-perfusion.

4) Lind GE, et al. (2017) published crystal structures of the GluN1/2A agonist binding domain heterodimer with bound ACEPC antagonists that reveal a binding mode in which the ligands occupy a cavity that extends toward the subunit interface between GluN1 and GluN2A ABDs. Can authors speculate whether this cavity could be used to further improve specificity and/or affinity of UBP1700?

Authors' response: We have tried in modelling studies to add substituents to UBP1700 to contact the cavity described by Lind et al (2017), but were unable to find substituents that could

interact with that site. We expect that major structural changes to UBP1700 would be needed to make use of this cavity to further improve GluN2D selectivity and affinity.

5) Why did the authors use different assays to determine drug affinity for different ionotropic glutamate receptors? TEVC for GluN1/GluN2A-D and glutamate-stimulated Ca²⁺-influx assay for GluA1, GluK1, and GluK2?

Authors' response: UBP compounds were initially characterised on GluA1, GluK1 and GluK2 in a Ca²⁺ fluorescence assay rather than TEVC due to its higher throughput. This assay is more effective when testing a number of different compounds initially. We know that relative differences in potencies in this fast assay coincide well with those measured by TEVC, thus, we did not see any need to measure potency of UBP1700 on GluA1, GluK1, and GluK2 using TEVC.

6) Authors should comment on why racemic mixtures of drugs instead of pure enantiomers were used.

Authors' response: Racemates of UBP791, UBP1700-1702 were used to enable us to readily compare activities to those obtained with previous analogues of PPDA in structure-activity relationship studies. Single enantiomers would be hard to obtain by asymmetric synthesis and this would greatly increase the time needed to synthesise individual compounds. We have obtained the individual enantiomers of PPDA by crystallisation of a salt of PPDA with a single enantiomer of a chiral base (Irvine et al., 2012; Jespersen et al., 2014). However, there is no guarantee that the same method would work for UBP791 and UBP1700, which are both much more water soluble than PPDA. In addition, the low yields in the synthesis of UBP791 and UBP1700 meant that we had insufficient material to attempt resolution to obtain the individual enantiomers.

This point is also addressed in response to Reviewer #2 above.

7) Authors couldn't crystalize GluN2D competitive antagonist bound LBD, so x-ray results are done on GluN2A-4m construct, which is GluN2A ligand-binding domain that mimics the GluN2D structure by introducing four key amino acid mutations into the GluN2A subunit. Authors provide evidence that in agonist bound state GluN2A-4m and GluN2D show very similar structure, however, it is not clear to what extent is this also valid for antagonist bound state. This should be addressed.

[Redacted]

8) Authors should add hill coefficient values into the dose-response plots.

Authors' response: We have added Hill coefficient values (n_H) to all dose-response plots.

Minor points:

1) Citation format on page 5 should be changed "(Furukawa & Gouaux, 2003; Furukawa et al, 2005; Karakas et al, 2011)"

2) glutamate, L-glutamate, L-Glutamate, L-glutamate; and Parkinson's disease vs Parkinson's Disease should be named consistently

3) In figure 3d, "(12)" shouldn't be inside of the plot
Unify

Authors' response: All fixed. Thank you for noticing these mistakes.

References

Brothwell, S. L. C., Barber, J. L., Monaghan, D. T., Jane, D. E., Gibb, A. J. & Jones, S. NR2B- and NR2D-containing synaptic NMDA receptors in developing rat substantia nigra *pars compacta* dopaminergic neurones. *J. Physiol.* **586.3**, 739-750 (2008).

Irvine, M. W., Costa, B. M., Dlaboga, D., Culley, G., Hulse, R., Scholefield, C. J., Atlason, P., Fang, G., Eaves, R., Morley, R., Mayo-Martin, M. B., Amici, M., Bortolotto, Z. A., Donaldson, L., Collingridge, G. L., Molnár, E., Monaghan, D. T. & Jane, D. E. Piperazine-2,3-dicarboxylic acid derivatives as dual antagonists of NMDA and GluK1-containing kainate receptors. *J. Med. Chem.* **55**, 327-341 (2012).

Jespersen, A., Tajima, N., Fernandez-Cuervo, G., Garnier-Amblard, E. C. & Furukawa, H. Structural Insights into Competitive Antagonism in NMDA Receptors. *Neuron* **81**, 366-378 (2014).

Jullienne, A., Montagne, A., Orset, C., Lesept, F., Jane, D. E., Monaghan, D. T., Maubert, E., Vivien, D. & Ali, C. Selective inhibition of GluN2D-containing N-methyl-D-aspartate receptors prevents tissue plasminogen activator-promoted neurotoxicity both in vitro and in vivo. *Molecular Neurodegeneration* **6**, 68 (2011).

Suárez, F., Zhao, Q., Monaghan, D. T., Jane, D. E., Jones, S. & Gibb, A. J. Functional heterogeneity of NMDA receptors in rat substantia nigra *pars compacta* and reticulate neurones. *Eur. J. Neurosci.* **32**, 359-367 (2010).

Volianskis, V., Bannister, N., Collett, V. J., Irvine, M. W., Monaghan, D. T., Fitzjohn, S., Jensen, M. S., Jane, D. E. & Collingridge, G. L. Different NMDA receptor subtypes mediate induction

of long-term potentiation and two forms of short-term potentiation at CA1 synapses in rat hippocampus *in vitro*. *J. Physiol.* **591**, 955-972 (2013).

Reviewers' Comments:

Reviewer #1:

Remarks to the Author:

The authors have changed the manuscript according to the suggestions of the reviewers. All suggestions were considered. The authors discussed all comments carefully.

Acceptance of the manuscript in the present form is strongly recommended.

Reviewer #2:

Remarks to the Author:

Thank you for addressing my comments. It is an excellent paper.

Reviewer #3:

Remarks to the Author:

The authors have adequately addressed all of the comments and improved their manuscript accordingly. The paper is very nice and deserving of acceptance.